



A Wavelet-Based Approach to Streamflow Event Identification and Modeled Timing Error
Evaluation
Erin Towler[1,*] and James L. McCreight[1,2]
[1] National Center for Atmospheric Research (NCAR), P.O. Box 3000, Boulder, CO 80307
[*] Corresponding author: towler@ucar.edu, https://orcid.org/0000-0002-1784-1346
[2] orcidid: 0000-0001-6018-425X
**Abstract**
Streamflow timing errors (in the units of time) are rarely explicitly evaluated, but are
useful for model evaluation and development. Wavelet-based approaches have been shown to
reliably quantify timing errors in streamflow simulations, but have not been applied in a
systematic way that is suitable for model evaluation. This paper provides a step-by-step
methodology that objectively identifies events, and then estimates timing errors for those events,
in a way that can be applied to large-sample, high-resolution predictions. Step 1 applies the
wavelet transform to the observations, and uses statistical significance to identify observed
events. Step 2 utilizes the cross-wavelet transform to calculate the timing errors for the events
identified in Step 1. The approach also includes a quantification of the confidence in the timing
error estimates. The methodology is illustrated using real and simulated stream discharge data
from several locations to highlight key method features. The method groups event timing errors
by dominant timescales, which can be used to identify the potential processes contributing to the
timing errors and the associated model development needs.  For instance, timing errors that are
associated with the diurnal melt cycle are identified. The method is also useful for documenting
and evaluating model performance in terms of defined standards. This is illustrated by showing
version-over-version performance of the National Water Model (NWM) in terms of timing
errors.
**1. Introduction**





Common verification metrics used to evaluate streamflow simulations are typically
aggregated measures of model performance, e.g., the Nash Sutcliffe Efficiency (NSE) and the
related root mean square error (RMSE). Although typically used to assess errors in amplitude,
these statistical metrics include contributions from errors in both amplitude and timing (Ehret
and Zehe 2011), making them difficult to use for diagnostic model evaluation (Gupta et al.
2008). Furthermore, common verification metrics are calculated using the entire time series,
whereas timing errors require comparing localized features or events in the data. This paper
focuses explicitly on event timing error estimation, which is not routinely evaluated, despite its
potential benefit for model diagnostics (Gupta et al. 2008) and practical forecast guidance (Liu et
al. 2011).
The fundamental challenge with evaluating timing errors is identifying what constitutes
as an "event" in the two time series being compared. Identifying events is typically subjective,
time consuming, and not practical for large-sample hydrological applications (Gupta et al. 2014).
Most methods for identifying events have focused on flooding events. One common approach to
identifying flooding events is to use peak-over-threshold methods. The thresholds used for such
analyses are often either based on historical percentiles (e.g., the 95[th] percentile) or on local
impact levels (river stage), such as the National Weather Service (NWS) flood
categories (NOAA National Weather Service, 2012). Timing error metrics are often calculated
from the peaks of these identified events. For example, the Peak Time Error, or its derivative the
Mean Absolute Peak Time Error, requires matching observed and simulated event peaks, and
calculating their offset (Ehret and Zehe 2011). While this may be straightforward visually, it can
be difficult to automate; some of the reasons for this are discussed below.



Difficulties arise using thresholds for event identification. For example, exceedances can
cluster if a hydrograph vacillates above and below a threshold, begging the question: Is it one or
multiple events? Which peak should be used for the assessment? In the statistics of extremes,
declustering approaches can be applied to extract independent peaks (e.g., Coles 2001), but this
reductionist approach may miss relevant features. For instance, if background flows are elevated
for a longer period of time before and after the occurrence of these "events", the threshold-based
analysis identifies features of the flow separately from the primary hydrologic process
responsible for the event. If one focuses just on peak timing differences in this example, that
timing error may only apply to some small fraction of the total flow of the larger event which
happens mainly below the threshold. Further, for overall model diagnosis that focuses on model
performance for all events, not just flood events, variable thresholds would be needed to account
for different kinds of events (e.g., a daily melt event versus a convective precipitation event).
Using a threshold-approach to identify events and timing error assessment, Ehret and
Zehe (2011) develop an intuitive assessment of hydrograph similarity, the Series Distance. This
algorithm is later improved upon by Siebert et al (2016). The procedure matches observed and
simulated segments (rise or recession) of an event, and then calculates the amplitude and timing
errors, as well as the frequency of event agreement. The Series Distance requires smoothing the
time series, identifying an event threshold, and selecting a time range to consider two segments
matching.
Liu et al (2011) developed a wavelet-based method for estimating model timing errors.
Although wavelets have been applied in many hydrologic applications such as model analysis
(e.g. Lane 2007; Weedon et al. 2015; Schaefli and Zehe 2009, Rathinasamy et al. 2014) and
post-processing (Bogner and Kalas 2007; Bogner and Pappenberger 2011), Liu et al. were the



first to use it for timing error estimation. Liu et al. (2011) apply a cross-wavelet transform
technique to streamflow time series for 11 headwater basins in Texas. Timing errors are
estimated for medium- to-high flow "events" that are determined a priori by threshold
exceedance. They use synthetic as well as real streamflow simulations to test the utility of the
approach. They show that the technique can reliably estimate timing errors, though they
conclude that it is less reliable for multi-peak or consecutive "events" (defined qualitatively).
ElSaadani and Krajewski (2017) followed the cross-wavelet approach used by Liu et al (2011) to
provide similar analysis and further investigate the effect of the choice of mother wavelet on the
timing error analysis. Ultimately, they recommended that in the situation of multiple, adjoining
flow peaks the improved time localization of the Paul wavelet might justify its poorer frequency
localization compared the Morlet wavelet.

Liu et al. (2011) provide a starting point for the work in this paper where we develop two

new bases for their method: 1) objective event identification for timing error evaluation and 2)
the use of observed events as the basis for the model timing error calculations The latter is
important for "model benchmarking", i.e., the practice of evaluating models in terms of defined
standards (e.g., Luo, et al. 2012; Newman et al. 2017). Here, the use of observed events provides
a baseline by which to evaluate changes and to compare multiple versions or experimental
designs.

This paper provides a methodology for using wavelet analysis to quantify timing errors in

hydrologic simulations. Our contribution is a systematic approach that integrates 1) statistical
significance to identify events with 2) a basis for timing error calculations independent of model
simulations (i.e., benchmarking). We apply our method to evaluation of high-resolution
streamflow prediction. The paper is organized as follows: Section 2 provides an overview of the



conceptual approach of using wavelets to identify events and estimate timing errors, and Section
3 provides the detailed methodology. In Section 4, we describe the software and data, as well as
provide a simple illustration of the method using real and simulated streamflow data. In Section
5, we provide results, including select examples to highlight features of the method and version-
over-version comparisons. Section 6 is the discussion and conclusions, including how specific
choices may vary by application.
**2. Conceptual Overview**

Before going into technical details of the Method (Section 3), we provide a conceptual

overview of the approach of using wavelets to identify events and estimate timing errors. We
provide a nomenclature table (Supplemental Table 1) of key terms relevant to the approach. The
wavelet transform (WT) expands the dimensionality of the original time series by introducing the
timescale (or period) dimension and returns power as a function of both time and timescale (e.g.
Torrence and Compo, 1998). This is illustrated in Figure 1: the streamflow time series (panel a) is
expanded into a 2-dimensional wavelet power spectrum (panel b). Where traditional model errors,
such as the aforementioned RMSE or NSE, reduce the information of the time series to a single
statistic, wavelet analysis expands the input signal and provides information on the dominant
timescales of the time series at each time. Wavelet analysis can therefore detect localized signals
in time series (Daubechies 1990), including hydrologic time series, which are often irregular or
aperiodic (i.e., events may be isolated and don't regularly repeat) or non-stationary. We note that
in many wavelet applications, timescale is referred to as "period". To emphasize that our study is
more focused on irregular events and less on periodic behavior of time series, we use the term
"timescale". The wavelet transform is the foundation of the view in this paper that events have





characteristics of both time and timescale. Timing errors, calculated from events defined this way,
therefore have dimensions of both time and timescale as well.

In their seminal wavelet study, Torrence and Compo (1998) outline a method for

objectively identifying statistical significance in the wavelet transform. We adopt this approach
and define "events" in the observed time series via statistical significance of the wavelet power
spectrum. The details are provided in the next section, however Figure 1 illustrates that the
events in the input time series (panel a) are defined as regions of the wavelet power spectrum
shown in panel b: events are inside the black contours (>= 95% confidence level) but not inside
the cone of influence (regions where the colors are muted, this is explained in detail in Section
3). The wavelet power spectrum is only shown for the events in panel c. Events defined in this
way are a function of both time and timescale. Note that at a given time, events of different
timescales can occur simultaneously. What one may subjectively interpret as a single event in the
input time series is generally quantified by this definition as multiple coincident events at a
variety of timescales each with a different power (e.g. Figure 1, panel c). Although for some
locations there may be physical reasons to expect certain timescales to be important (e.g.,
seasonal cycle of snowmelt), the most important scales at which hydrologic signals occur at a
particular location are not necessarily known a priori. The wavelet power can be examined
across events to identify the most dominant, or what we call "characteristic" timescales for a
given time series; the procedure for this is detailed later in the technical methodological section
(Section 3.1.3). This approach to event detection is objective, data-driven, and portable across
diverse locations, which is important for large-sample hydrologic applications. We point out that
in the objective identification of events, we are not limited to flooding events. Rather, events are





defined more broadly: an event is when the wavelet power falls outside its standard statistical
power. This can be further subset into flooding events if desired.

Once observed events are identified by the method, we can calculate timing errors

between observed and simulated time series. The cross-wavelet timing error approach of Liu et
al (2011) is used, but we restrict our calculation of timing errors to the aforementioned regions of
statistically significant wavelet power in the observations; i.e., we calculate timing errors in
terms of *observed* events (Figure 1c). Because both the phase (timing error) and the significance
of the cross wavelet transform (XWT) computed between the observed and modeled time series
depends on the modeled time series, we use the observed event definition (Figure 1c) in the
calculation of the timing errors to provide a common, consistent basis independent of the models
evaluated (i.e., benchmarking). The portions of the observed wavelet spectrum used for
comparison may further be restricted depending on the analysis goals.
**3. Method for evaluating event timing errors**
This section provides the technical description of the methodology, and the steps can be

seen in an accompanying flowchart (Supplemental Figure 1).
*3.1. Step 1. Identify observed events*
The first step towards evaluating timing errors is to identify a set of observed events for

which the timing error should be calculated. We break this step into three sub-steps: 1a. Apply
the wavelet transform to observations, 1b. Determine all observed events using significance
testing, and 1c. Sample observed events to an event-set relevant to analysis.
3.1.1. Step 1a. Apply wavelet transform to observations

First, we apply the continuous wavelet transform to the observed time series. We provide

an overview of the main steps and equations for the wavelet transform here, though the reader is
referred to Torrence and Compo (1998) and Liu et al. (2011) for more details.



Before applying the WT, a mother wavelet needs to be selected. In Torrence and Compo
(1998), they discuss the key factors that should be considered when choosing the mother
wavelet. There are four main considerations, including (i) orthogonal or nonorthogonal, (ii)
complex or real, (iii) width, and (iv) shape. In this study, we follow Liu et al. (2011) in selecting
the nonorthogonal and complex Morlet wavelet:
$$\psi(n) = \pi^{-1/4} e^{i w_0 n} e^{-n^2/2},$$
where $w_0$ is the non-dimensional frequency, with a value of 6 (Torrence and Compo, 1998).
Once the mother wavelet is selected, the WT is applied to a time series $x_n$, where n goes
from n=0 to n=N-1, with a time step of $\delta t$. The WT is the convolution of the time series with the
mother wavelet that has been scaled and normalized:
$$W_n(s) = \sum_{n'=0}^{N-1} x_{n'} \psi^* \left[ \frac{(n'-n)\delta t}{s} \right],$$
where s is the scale parameter, the asterix indicates the complex conjugate of the wavelet
function. The wavelet power is defined as $|W_n^2|$. We use the bias corrected wavelet power
spectrum (Liu et al. 2007; Veleda et al. 2012), which ensures spectral peaks are comparable
across timescales. We also identify a maximum timescale that corresponds to our application.
We select 256 hours (~10 days), but this number could be higher or lower for other applications
and there are no real penalties for using too high a maximum (lower than the annual cycle).
Because we are applying the WT to a finite time series, there are timescale-dependent
errors at the beginning and end times of the power spectrum. These are referred to as the cone of
influence or COI (Torrence and Compo, 1998). We ignore all results within the COI in this
study.
3.1.2. Step 1b. Determine all observed events using significant testing
Once the WT is applied, the 2-dimensional (2-D) wavelet power spectra shows how the
features of the time series vary with both time and timescale. To identify areas of significance,
we apply Torrence and Compo's (1998) approach that compares the WT power spectra with a
power spectra from a red noise process. Specifically, the observed time series is fitted with an
order 1 autoregressive (AR1, or red noise) model, and the WT is applied to the AR1 time series.
The power spectra of the AR1 model provide the basis for the statistical significance testing.
Significance is determined if the power spectra are statistically different using a chi-squared test
with 95% confidence.

Statistical significance indicates an "event" at a given time and timescale: that is, the

wavelet power falls outside its standard statistical power. The result is the set of all events, i.e.,
each event is a combination of time and timescale (i.e., locations on the 2-D grid).  We refer to
contiguous regions of statistical significance (in time and timescale) as "event clusters" (note that
no statistical clustering is performed).
3.1.3. Step 1c. Sample observed events to an event-set relevant to analysis

Step 1b results in the identification of all events at all timescales and times. In this sub-

step, the event space is sampled to suit the particular evaluation. Because the goal of this paper is
to evaluate model timing errors over long simulation periods, we choose to sample the event
space based on dominant timescales in the time-averaged observed wavelet spectra. For our
application we choose to further sub-sample the observed wavelet spectra by selecting, for each
characteristic timescale, the most powerful event within each event cluster. This is articulated in
the following bullets:
•   *Calculate the average event power across each timescale:* Considering only the

statistically significant areas of the observed wavelet spectrum, calculate the average

power across each timescale.




• *Identify timescales of absolute and local average power maxima:* By plotting the average

power versus the timescale, the local and absolute maximums for average power can be

determined. The timescales corresponding to the absolute and local maxima of the

average power are called the characteristic timescales of the observed wavelet spectrum.

This is the first subset of events: all events that fall within the characteristic time scales.

• *Identify events with maximum power for each event cluster:* As previously mentioned,

events can also be grouped into "event clusters", that is, contiguous significant areas. We

can use this to further sample from the event-set created in the last bullet: across each

characteristic timescale, we identify the event with maximum power for each event

cluster. This is the second event subset: all events with maximum power for each cluster

that fall within a characteristic timescale.

*3.2. Step 2. Calculate Timing Errors*

Step 1 identifies events by applying a wavelet transform to the observed time series. To

calculate the timing error of a modeled time series, we perform its cross wavelet transform with
the observed time series, as detailed in this section.
3.2.1. Step 2a. Apply cross-wavelet transform (XWT) to observations and simulations

Given the WT of an observed time series $W_n^X(s)$ and a modeled time series $W_n^Y(s)$, the

cross-wavelet spectrum can be defined as:

$$W_n^{XY}(s) = W_n^X(s)W_n^{Y*}(s),$$

where the asterix implies the complex conjugate. The cross-wavelet power is defined as
$|W_n^{XY}(s)|$.

Similar to Step 1b of the WT, we can also calculate the areas of significance for the

XWT. These are not the same as the areas of significance for the WT. The significant areas of





the XWT vary with each simulation, and are therefore not useful for evaluation on their own.
Nevertheless, we are interested in the overlap between the significant areas of the observed WT
and the significant areas of the cross-wavelet transform, and this is used to quantify our
confidence in the timing error estimate. We discuss this further in Step 2d.
3.2.2. Step 2b. Calculate the cross-wavelet timing errors

To calculate the timing errors, we first compute the phase angle of the cross-wavelet

spectrum. The phase angle gives the phase difference and can be computed as:
$$\phi_n^{XY}(s) = tan^{-1}\left[\frac{\Im(\langle s^{-1}W_n^{XY}(s)\rangle)}{\Re(\langle s^{-1}W_n^{XY}(s)\rangle)}\right],$$

where $\Im$ is the imaginary and $\Re$ is the real component of $W_n^{XY}(s)$.

We convert the phase angle into the timing error as in Liu et al. (2011):

$$\Delta t_n^{XY}(s) = \phi_n^{XY}(s) * T/2\pi,$$

where T is the equivalent Fourier period of the wavelet.
3.2.3. Step 2c. Subset cross-wavelet timing errors to sampled observed events

Step 2b results in an estimate of timing errors for all times and timescales in the cross-

wavelet transform space. In our application we are interested in the timing errors that correspond
to the identified sample of *observed* events, especially for events at the characteristic timescales
(the first event-set in step 1c) and for the maximum power events in each cluster (the second
event-set in step 1c). The latter provides a single timing error for each event cluster at each
characteristic timescale, which could be used in a post-processing step to provide a cluster-by-
cluster timing correction, if desired.

It is important to point out that for other applications, there could be other ways to

interrogate the timing errors that result from the cross-wavelet transform. Some of these
possibilities are noted in the Discussion section.
3.2.4. Step 2d. Quantify the confidence in the timing error estimate

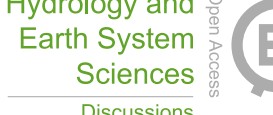

To interpret our confidence in the timing error estimate, we can examine the overlap
between the significant areas of the observed WT and the significant areas of the XWT.
We can look at percent (%) overlap, that is, how many of the XWT events overlap with
the WT events, either for all events or for the sampled event-sets. An overlap close to 0% would
indicate that the model did not do a good job of simulating the observations – or it is a "miss"
(flood is observed but not forecasted). If the overlap was 100%, it would be close to a perfect
simulation. Second, if we are looking at a single timing error for each event cluster, we may look
to see if that event is significant in the XWT. If it is not, it gives us less confidence in the
estimate.
We note that because we are calculating timing errors in terms of observed events, there
is no information about "false alarms", where a flood is forecasted but not observed.
**4. Application of the Framework**
The methodology developed in this paper is implemented in the R language and is made
publicly available, as detailed in the code availability section at the end of the manuscript.
*4.1. Data*
The application of the methodology is illustrated using real and simulated stream discharge
(streamflow, m3/s) data from four U.S. Geological Survey (USGS) stream gage locations: Onion
Creek at US Highway 183, Austin, Texas (Onion Creek, TX; USGS site number 08159000),
Taylor River at Taylor Park, Colorado (Taylor River, CO; USGS site number 09107000),
Pemigewasset River at Woodstock, New Hampshire (Pemigewasset River, NH; USGS site
number 01075000), and Bad River near Fort Pierre, South Dakota (Bad River, SD; USGS site
number 06441500).  We use the USGS instantaneous observations averaged on an hourly basis.
NOAA's National Water Model (NWM,
https://www.nco.ncep.noaa.gov/pmb/products/nwm/) is an operational model that produces



hydrologic analyses and forecasts over the continental United States (CONUS) and Hawaii (as of
version 2.0). The model is forced by downscaled atmospheric states and fluxes from NOAA's
operational weather models. Next, the NoahMP (Niu et al 2011) land surface model calculates
energy and water states and fluxes. Water fluxes propagate down the model chain through
overland and subsurface (soil and aquifer representations) water routing schemes to reach a
stream channel model. The NWM applies the three parameter Muskingum-Cunge river routing
scheme to a modified version of the NHD-Plus version 2 (McKay et al. 2012) river network
representation.
In this study, NWM simulations are taken from each version's retrospective runs
(https://docs.opendata.aws/nwm-archive/readme.html). These are continuous simulations (not
cycles) run for the period October 2010 to November 2016 and forced by the National Data
Assimilation System (NLDAS)-2 product as atmospheric conditions. The nudging data
assimilation was not applied in these runs either. We use NWM discharge simulations from
versions V1.0, V1.1, and V1.2 (not all version may be publicly available).

To apply the methodology, we note that the observed and simulated datasets must be

paired (overlapping). Further, for evaluation, any new simulation must also be paired with the
observed. Missing data, which is common in observed time series, can be problematic and can
result in false significance. We account for this our methodology by calculating the XT and
XWT on each complete time series. This will be illustrated in the forthcoming example at Taylor
River, CO.
*4.2. Application*

For illustration purposes we apply Steps 1 and 2 to an observed time series in Onion

Creek, TX; for simplicity, we select an isolated peak (Figure 1a). First, we apply the wavelet
transform to the observations (Figure 1b). This shows the time series in terms of its power by
time and timescale, with warmer colors indicating more power. The black outline shows the
areas of significance and the muted colors indicate the COI. To determine all observed events,
we identify all the points that are significant and outside the COI (Figure 1c). Next, we average
the power across each timescale: to the right of Figure 1b we show power averaged across all
points for each timescale, and to the right of Figure 1c we show power averaged across just the
events for each timescale. The latter is the one used to identify our characteristic scales. In this
case, there is a single maximum at 22 hours. For the characteristic timescale, we see there is only
1 event cluster and the event with maximum power is marked with a star (Figure 1d).
For Step 2, we use the same Onion Creek, TX, peak from Figure 1a, and add a prescribed
timing error of +5 hours to every point in the original time series (Figure 2a) to create a synthetic
time series. We perform the cross-wavelet transform between the observed and synthetic time
series (Figure 2b). The arrows in Figure 2b indicate the phase offset, which are used to calculate
the timing error (Figure 2c). The timing error estimates show that for timescales greater than 10
hours, we get back the prescribed timing error of 5 hours, i.e., the scale must be at least double
the timing error. In this case, because we are adding a prescribed error, the error is approximately
5 hours for all events, including for the characteristic timescale of 22 hours.
Finally, we repeat Step 2, but compare the observation of this event to actual model data
from NWM V1.2. This shows that the model is early (Supplemental Figure 2a). We perform the
cross wavelet transform (Supplemental Figure 2b) and examine the timing error (Supplemental
Figure 2c). Table 1 summarizes the results: the mean error across the 22-hour characteristic
timescale is -3.2 hours, as is the error for the cluster's maximum power. All events in the cluster
are also significant in the XWT (100%), and the cluster maximum is also significant, providing
confidence in this timing error estimation.



**5. Results**
In this section, modeled data is used from several locations and time series to highlight the
features of the method, finishing with version-over-version comparisons to illustrate the utility
for evaluation.
5.1. Pemigewasset River, NH
This example uses time series from the Pemigewasset River, NH. First, we examine a three-
month time series that exhibits multiple peaks above a base flow (Figure 3a). By eye, it is fairly
straightforward to pick out three main peaks. The wavelet transform (Figure 3b and 3c) reveals
up to three event clusters, depending on the characteristic timescale examined (Figure 3d). When
we plot the average power by timescale (right of Figure 3c), we see that there are nine relative
maxima (small grey dots) – hence there are 9 characteristic scales for this example.
In Step 2, we compare the same time series with output from NWM V1.2 (Figure 4a), apply
the cross-wavelet transform (Figure 4b), and calculate the timing error for all observed events
(Figure 4c). As previously mentioned, we are interested in the timing errors corresponding to
observed events at the characteristic timescales. In Figure 5a, the panels are ordered by
timescales from highest to lowest average power; we only show the top 5 characteristic scales,
using the first-subset of events, grouped by cluster. The first panel, where timescale = 24.8 hours,
is the absolute maximum. This shows two cluster distributions: for cluster one, the model is late
for most events, and cluster two shows the model is early; the dark shading indicates that most of
the events are significant in the XWT. The next two dominant scales have similar average power
and are of the same order of magnitude at 27.8 hours and 33.1 hours; if we had applied
smoothing to the graph of average power by timescale, these relative maxima would smooth out.
We will revisit this in the Discussion, when we discuss pathways to implementation. The



characteristic scale with the next highest maxima occurs at 111 hours, which is a different order
of magnitude, suggesting that this may have a different physical process driving it. This shows
the model to be late for both clusters, and results are similar for a timescale of 148 hours. We
don't show results for the remaining 4 characteristic time scales with lower average power, since
they have similar characteristic timescale values and associated timing errors to what has already
been shown.

We can see how looking at the timing errors using the cluster distributions will get harder as

the number of clusters increase, so it is also useful to summarize the information by looking at
each cluster mean and max. If we run the methodology on the full 5-year Pemigewasset River
time series, we can compare the mean and max timing errors for each characteristic time scale
using box plots where the outline is shaded by the average confidence (Supplemental Figure 3).
Table 2 summarizes this information. For example, the absolute maxima, at the 17.5 hour
timescale has 86 clusters, and a timing error centered around zero (-0.43 hours), 75% of which
are significant in the XWT. This is very similar to the results for the cluster max, as it is for the
rest of the characteristic time scales. One other thing to note is that as expected, because the
characteristic time scales are data driven, they are not the same as they were for the 3-month
period.
5.2. Bad River, SD

The second example uses a two-month time series from the Bad River, SD, to illustrate the

concept of consecutive peaks (Figure 6a). Whereas in the previous example it was fairly
straightforward to pick out 3 distinct peaks, in this time series, there is one noticeable peak
centered around June the 1st, with smaller peaks preceding and following it. The question is
whether or not this is one event cluster or multiple? Looking at the wavelet transform (Figure 6b



and 6c), we can see that for smaller timescales, there are more clusters, but for longer timescales,
they are considered a single cluster.
In Step 2, we compare the same time series with output from NWM V1.2 (Figure 7a),
calculate the cross-wavelet transform (Figure 7b), and calculate the timing error (Figure 7c). The
timing error figure shows a sign switch: for longer timescales (i.e., when the peaks are
considered part of a single event cluster), the model is early, but for shorter time scales (i.e.,
when the peaks are each considered their own cluster), the model is late. This is an important
point: corrections at one scale may worsen timing error (or other metrics) at other scales.
This example has another interesting feature: namely that there is a false alarm in the model
just before July 15. We note that because of our methodology, there is no observed event at that
time, and therefore no timing error to be calculated, that is there is no information in the timing
error statistics in terms of false alarms.
5.3. Taylor River, CO
In this example, we will examine a time series from Taylor River, CO, that illustrates peaks
that are driven by different processes. The Taylor River is in a mountainous area where the
spring hydrology is dominated by snowmelt runoff. To start, we will look at a portion of the
spring melt season, where we can visibly see a diurnal signal (Figure 8). However, while it's
easy to see that the model is too high in amplitude, it's hard to visually tell much about the
timing error. Figure 9 shows that for the characteristic time scale of 23.4 hours, the model is
usually early, with high confidence.
Supplemental Figure 4a shows a year-long time series from Taylor River, CO, where we can
see the snowmelt runoff in spring, but also several peaks in summer, likely driven by summer
rains. From the WT, we again see the peak in the characteristic time scales at about 24 hours
(right of Supplemental Figure 4c), but there is another maxima at 99 and 118 hour timescales,



relating to flows from the summer rains. Looking at Figure 10, starting with the 24 hour
timescale, we see that for the clusters that are significant in the XWT, the model is generally
early. For the 118 and 99 hour timescale, the model is also early, but those cluster events are not
statistically significant in the XWT. This suggests that we are confident in the early timing errors
of the model for the diurnal snowmelt cycle, and this could be used as qualitative guidance for
model performance at this site until the model performance is improved. However, we show that
it is less reliable for the early timing errors for the summer peaks. This underscores the key point
that timing errors are timescale dependent, and can help diagnose which processes to target for
improvements.

Supplemental Figure 4b also illustrates how missing data is handled: this results in additional

COIs (muted colors) to account for the edge effects, and areas of the COI are ignored in our
analyses.
5.4. Evaluating Model Performance
Finally, we show how the methodology can be used for evaluating performance changes

across NWM versions. We point out that none of the NWM version upgrades were targeting
timing errors, so these results just provide a demonstration. We use a 5-year overlapping time
series and cluster max for the results, but cluster mean results were similar (not shown). For the
NWM V1.0 for Onion Creek, we see that for the 29.5 hour timescale, there were 17 clusters, for
which the median timing error is -1.4 hours, and all were significant in the XWT (Table 3).
Comparing V1.0, V1.1, and V1.2, the results for Onion Creek show that the median timing error
has gotten slightly earlier (worse), although the distribution became tighter from V1.0 to V1.1
and V1.2 (Figure 11). In Figure 11, the dark blue color of the boxplot outline indicates that there
is high confidence in the timing error, as the overlapping significance is close to 100% for the



top three characteristic timescales. Using the 5-year overlapping time series for Pemigewasset
River, NH, we see that the median timing error improved by getting closer to zero, but that the
distribution became wider (Figure 12). Again, the confidence is fairly high (>80%) across
characteristic time scales and versions (Table 4), and >60 clusters were used in the estimations.
Using 5-years from Taylor River, CO (Supplemental Table 2, Supplemental Figure 5), we see
that for the characteristic scale of 235 hours (~10 days), has low confidence (~25%) for the 4
sampled clusters; the timescale of 23.4 hours has a median that is consistently early by around 6
hours, with the version model confidence ranging from 44% to 67% (Supplemental Table 2).
Results for the Bad River can be seen in Supplemental Table 3 and Supplemental Figure 6.
**6. Discussion and Conclusions**
In this paper, we develop a systematic, data-driven methodology to objectively identify

events and estimate timing errors in large-sample, high-resolution hydrologic models. The
method was developed towards several intended uses: Primarily, it was developed for model
evaluation, so that model performance can be documented in terms of defined standards. We
illustrate this with the version-over-version NWM comparisons. Second, it can be used for model
development, whereby potential timing error sources can be diagnosed and targeted for
improvement. Related to this point, given the advantages of calibrating using multiple-criteria
(e.g., Gupta et al. 1998), timing errors could be used as part of a larger calibration strategy.
However, as noted in the consecutive peaks example for the Bad River, minimizing timing errors
at one timescale may not translate to improvements in timing errors (or other metrics) at other
scales. Wavelet analysis has also been used directly as an objective function for calibration,
although a difficulty is in determining the similarity measure to use (e.g. Schaefli and Zehe 2009,
Rathinasamy et al. 2014). Future research will investigate the properties of timing errors for



calibration. Finally, the approach can be used for model interpretation, as estimating timing
errors provides a characterization of the uncertainty (i.e., for a given timescale, the model is
generally late or early), as well as a measure of the confidence, that could be useful for
qualitative forecast guidance.

Given the fact that several subjective choices were made specific to our application and

goals, we think it is important to highlight that we have made the analysis framework openly
available (detailed in the code availability section below), so the method can be extended or
refined by the community right away. For instance, because of our focus on model evaluation
and development, we use the observed WT to identify events. However, in other instances it
might be sufficient to only sample events that are in the significant areas of the XWT (essentially
to identify the characteristic scales and event-set directly from the XWT instead of from the
WT). This might be reasonable for applications that are more focused on model interpretation in
a real-time forecasting mode, but it would not allow for version comparison and it is not
guaranteed that all the important characteristic scales would be identified (i.e., the model may
not capture some real-world processes, and therefore miss the associated characteristic
timescales).  We only look at the timing errors from an event-set relevant to our analysis, but
there are other ways to subset the events that might be more suitable to other applications. For
instance, we define the event set broadly, but it could be subset for high peak or flooding events
to compare with traditional peak-over-threshold approaches. For example, Supplemental Figure
7 shows the maximum streamflows for the event-set from the 5 year run at Taylor River; this
event-set could be filtered to include only events above a given threshold. The method provides a
quantification of the confidence in the timing errors, and we include all timing errors in our
summaries. However, it might make more sense to drop points that do not have a high



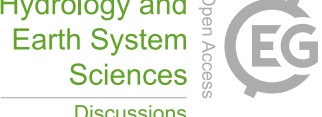

confidence (i.e., with a low percent of events that significantly overlap between the XT and the
XWT) and to flag those events as misses.

Another point that arises is how many characteristic timescales should be examined.

Here, we average the power across timescales and identify characteristic scales to be at every
absolute and relative maxima. As seen in the illustrative examples, this can result in multiple
characteristic scales, some of which can be quite similar, suggesting that events at those scales
are from similar or related processes. One solution could be to smooth the average power by
timescale, which would reduce the number of local maxima, or to look at timing errors within a
band of timescales. It is also important to note that the characteristic scales are data-driven, so
they will change with different lengths of observed time series. Longer runs capture more events
and should converge on the more dominant timescales and events for a location. However, for
performance evaluation, overlapping time periods are needed.

In our application of the WT, we follow Liu et al. (2011) and select the Morlet as the

mother wavelet. However, results are sensitive to the mother wavelet selected. Further discussion
of mother wavelet choices can be found in Torrence and Compo (1998) and in ElSaadani and
Krajewski (2017).

In short, this paper provides a systematic, flexible, and computationally efficient

methodology that is appropriate for model evaluation and comparison, and is useful for model
development and guidance. Future work will apply the approach to identify characteristic
timescales across the United States, as well as to assess the associated timing errors in the NWM.
**Code/Data Availability**
The code for reproducing the figures in this paper as well as extended vignettes/notebooks are
provided in public github repository https://github.com/NCAR/wavelet_timing. In addition to





reproducing the analyses and figures in this paper, several jupyter notebooks provide more
detailed analyses of the time series included in this paper. We emphasize that the analysis
framework is meant to be flexible and adapted to similar applications where different statistics
may be desired. The figures created are specific to the applications in this paper but provide a
starting point for other work.
The core code is provided in the public "rwrfhydro" R package
https://github.com/NCAR/rwrfhydro. The package can be installed as described by the
README document in the repository and in the Supplemental Online Materials for this paper.
The code is written in the open-source R language (R Core Team 2019) and builds off multiple,
existing R packages. Most notably the wavelet and cross-wavelet analyses are performed using
the "biwavelet" package (Gouhier et al. 2018).
**Credit Author Statement**
ET and JLM collaborated to develop the methodology.  ET led the results analysis and
manuscript preparation and revisions. JLM developed the initial idea for the work, the open
source software, and visualizations.
**Competing interests**. The authors declare that they have no conflict of interest.
**Acknowledgements**:

The authors would like to thank Dave Gochis for useful discussions and Aubrey Dugger

for providing NWM data. We thank the NOAA/OWP and NCAR NWM team for its support of
this research. This research is funded by the NOAA Office of Water Prediction and the Joint
Technology Transfer Initiative grant 2018-0303-1556911. This material is based upon work
supported by the National Center for Atmospheric Research (NCAR), which is a major facility





sponsored by the National Science Foundation (NSF) under Cooperative Agreement No.

1852977.

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



621                                              Figures



Figure 1. An isolated peak from Onion Creek, TX: (a) observed time series, (b) observed wavelet
power spectrum (left) and average power by timescale for all points (right); (c) statistically
significant wavelet power spectrum or events (left) and average power by time scale for all
events with maxima shown by grey dots (right); (d) Characteristic scale event cluster (horizontal
green line) and maxima (star).



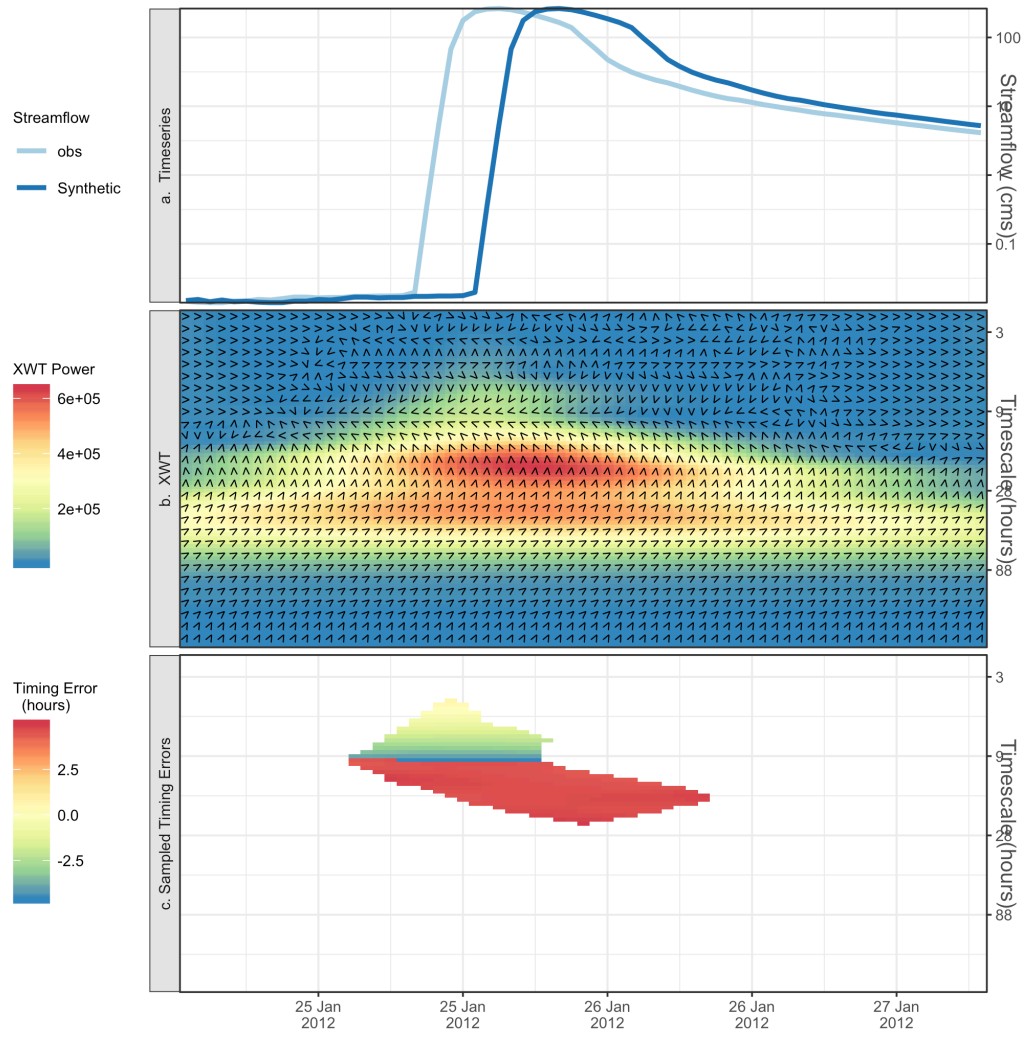

Figure 2. An isolated peak from Onion Creek, TX and a synthetic +5 hour offset: (a) observed
and synthetic time series, (b) cross wavelet (XWT) power spectrum and phase angles (arrows),
(c) sampled timing errors for observed events.



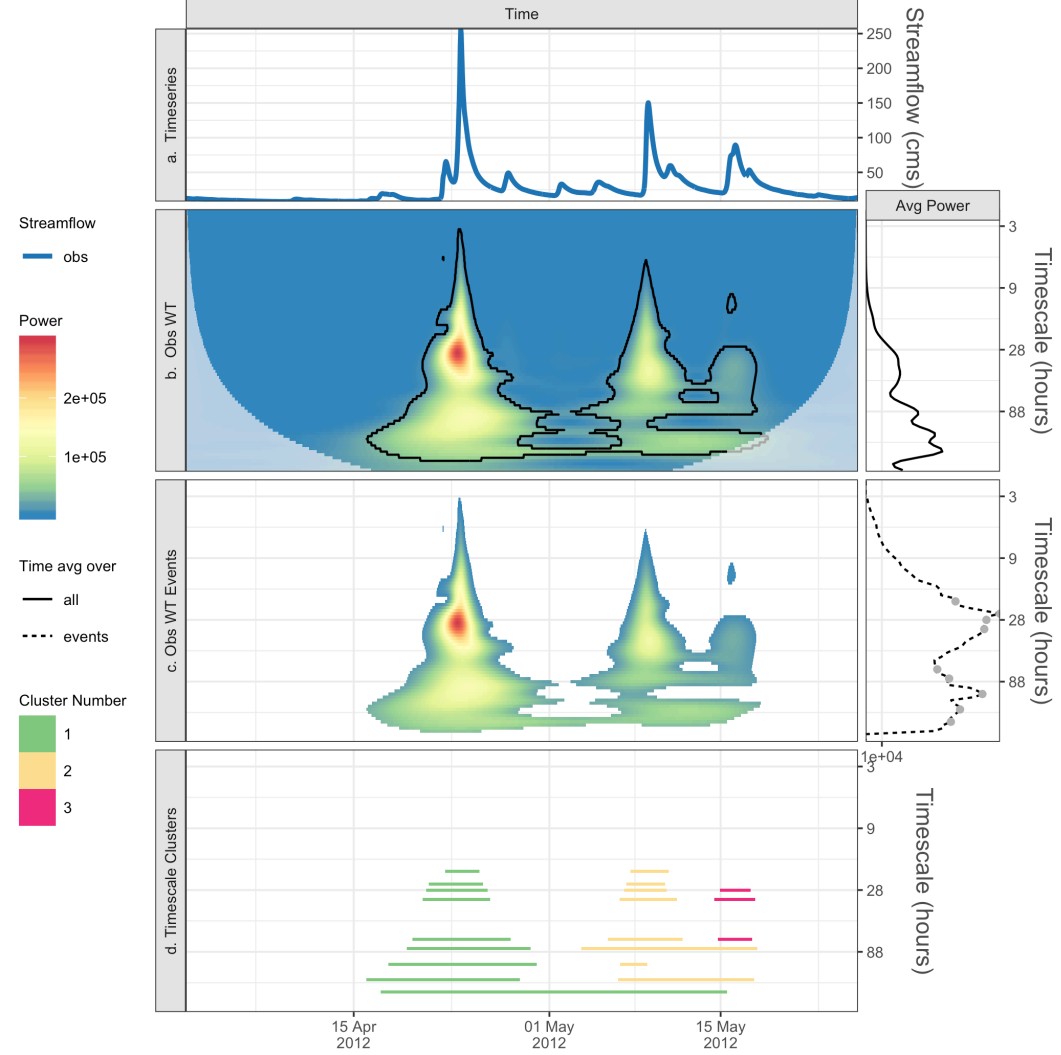

Figure 3. Multiple peaks from Pemigewasset River, NH: (a) observed time series, (b) observed wavelet power spectrum (left) and average power by timescale for all points (right); (c) statistically significant wavelet power spectrum or events (left) and average power by time scale for all events with maxima shown by grey dots (right); (d) Characteristic scales event clusters (horizontal lines).



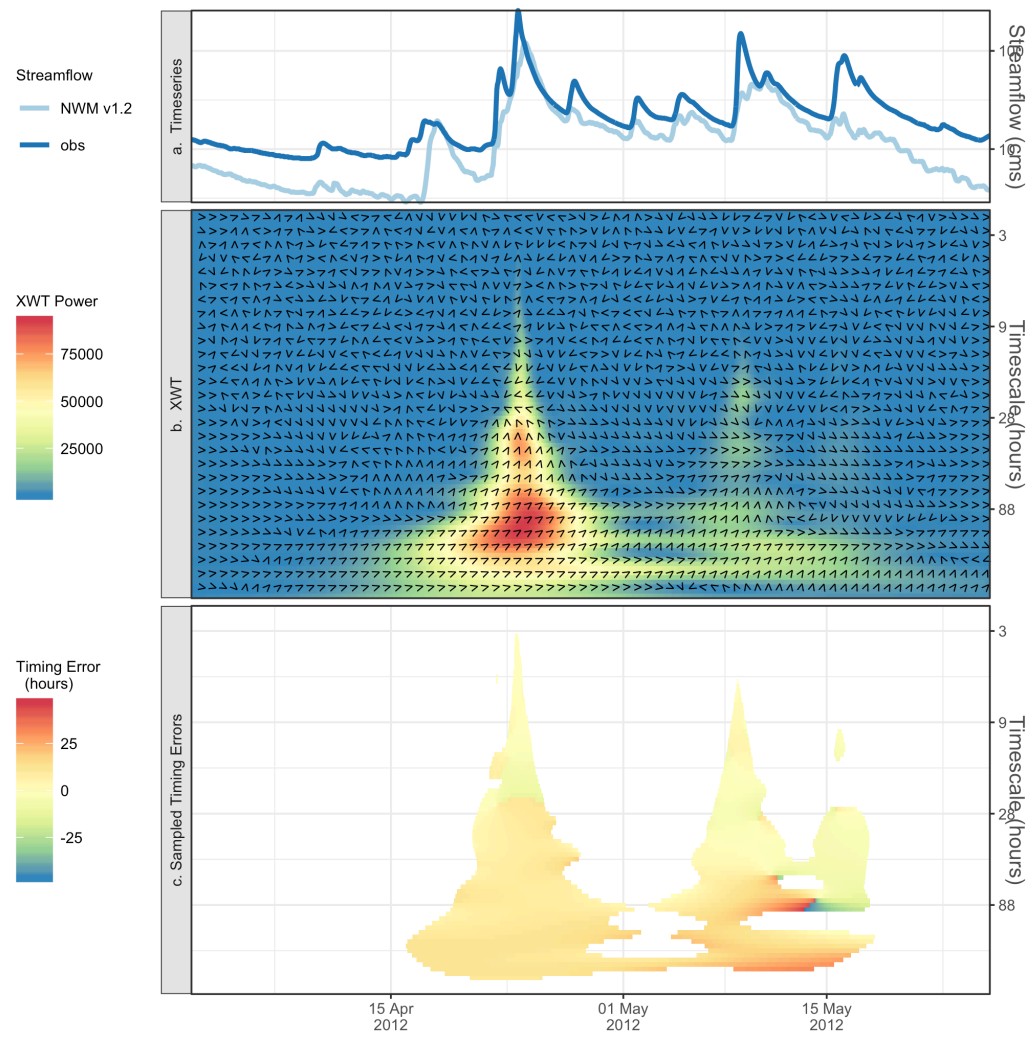

Figure 4. Multiple peaks from Pemigewasset River, NH: (a) observed and simulated NWM time
series, (b) cross wavelet (XWT) power spectrum and phase angles (arrows), (c) sampled timing
errors for observed events.





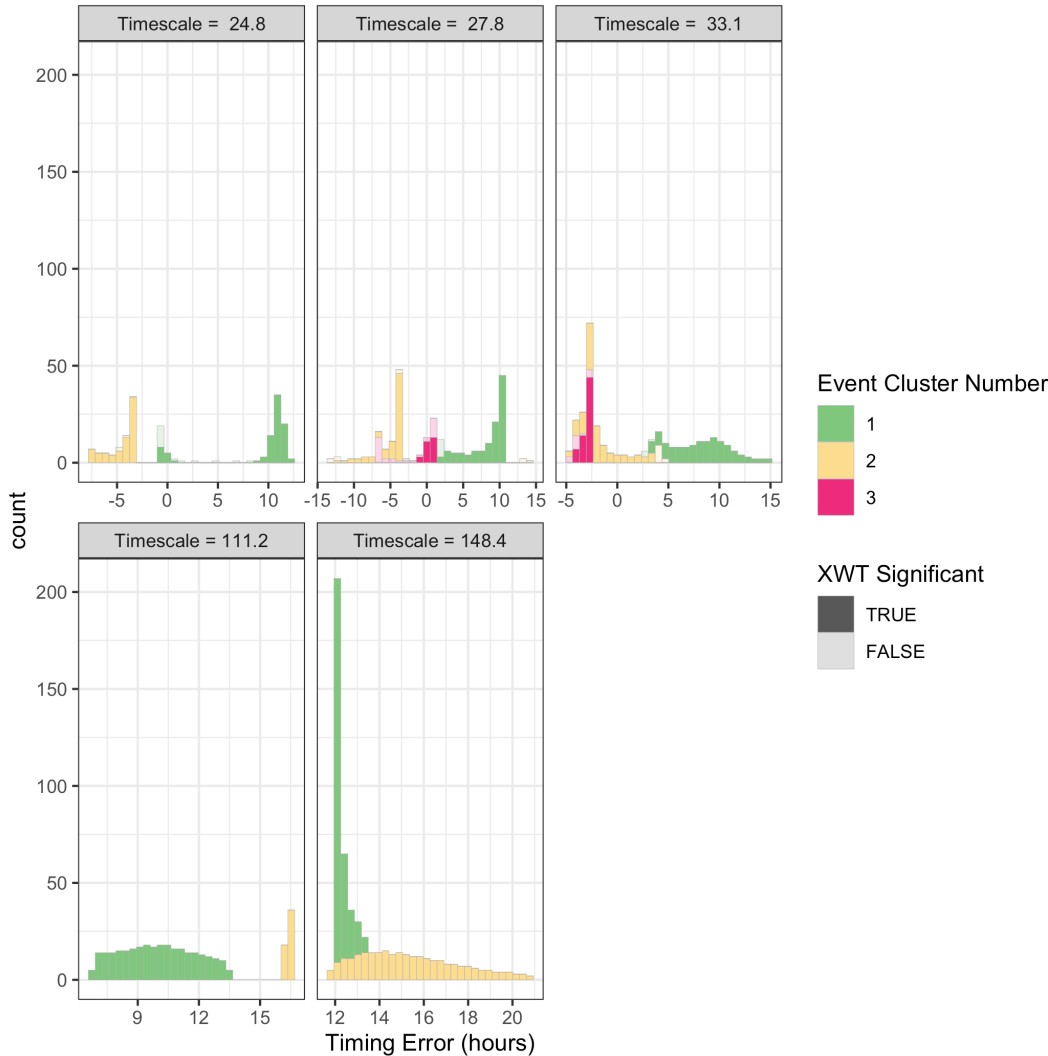

Figure 5. Multiple peaks from Pemigewasset River, NH : For the top 5 characteristic timescales (see panel title), timing error distributions for event clusters. Dark colors show if the event was significant in the cross wavelet transform (XWT), muted colors indicate no significance.



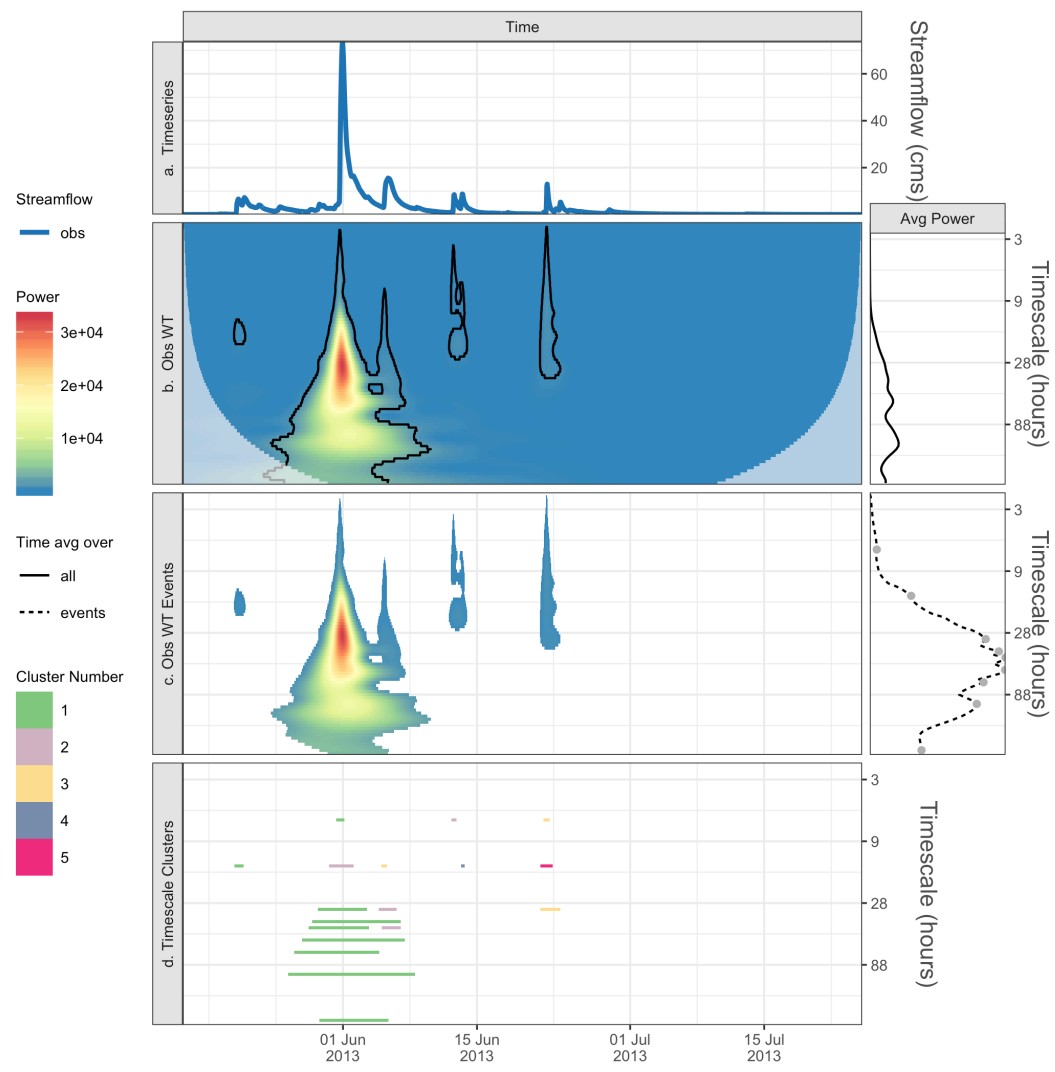

Figure 6. Consecutive peaks from Bad River, SD: (a) observed time series, (b) observed wavelet
power spectrum (left) and average power by timescale for all points (right); (c) statistically
significant wavelet power spectrum or events (left) and average power by time scale for all
events with maxima shown by grey dots (right); (d) Characteristic scales event clusters
(horizontal lines).

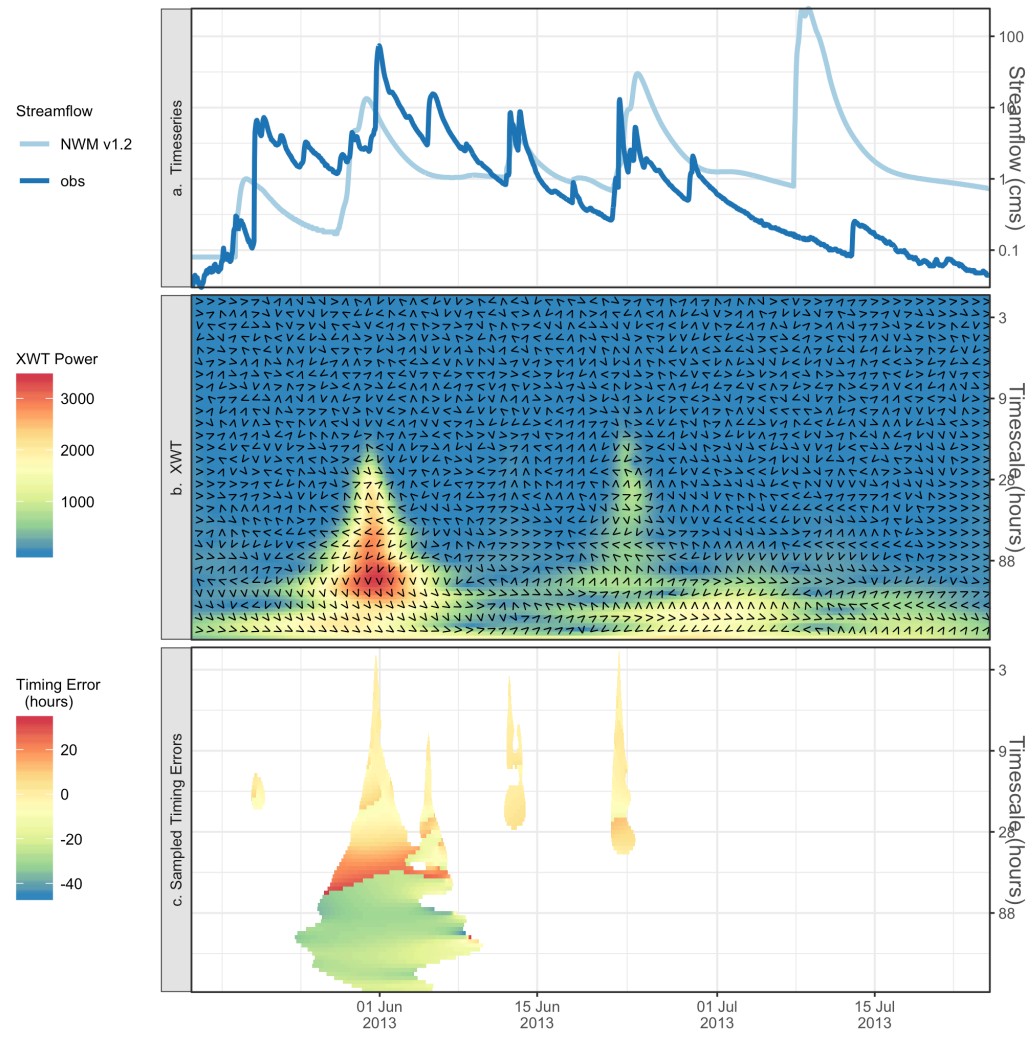

Figure 7. Consecutive peaks from Bad River, SD: (a) observed and simulated NWM time series,
(b) cross wavelet (XWT) power spectrum and phase angles (arrows), (c) sampled timing errors
for observed events.





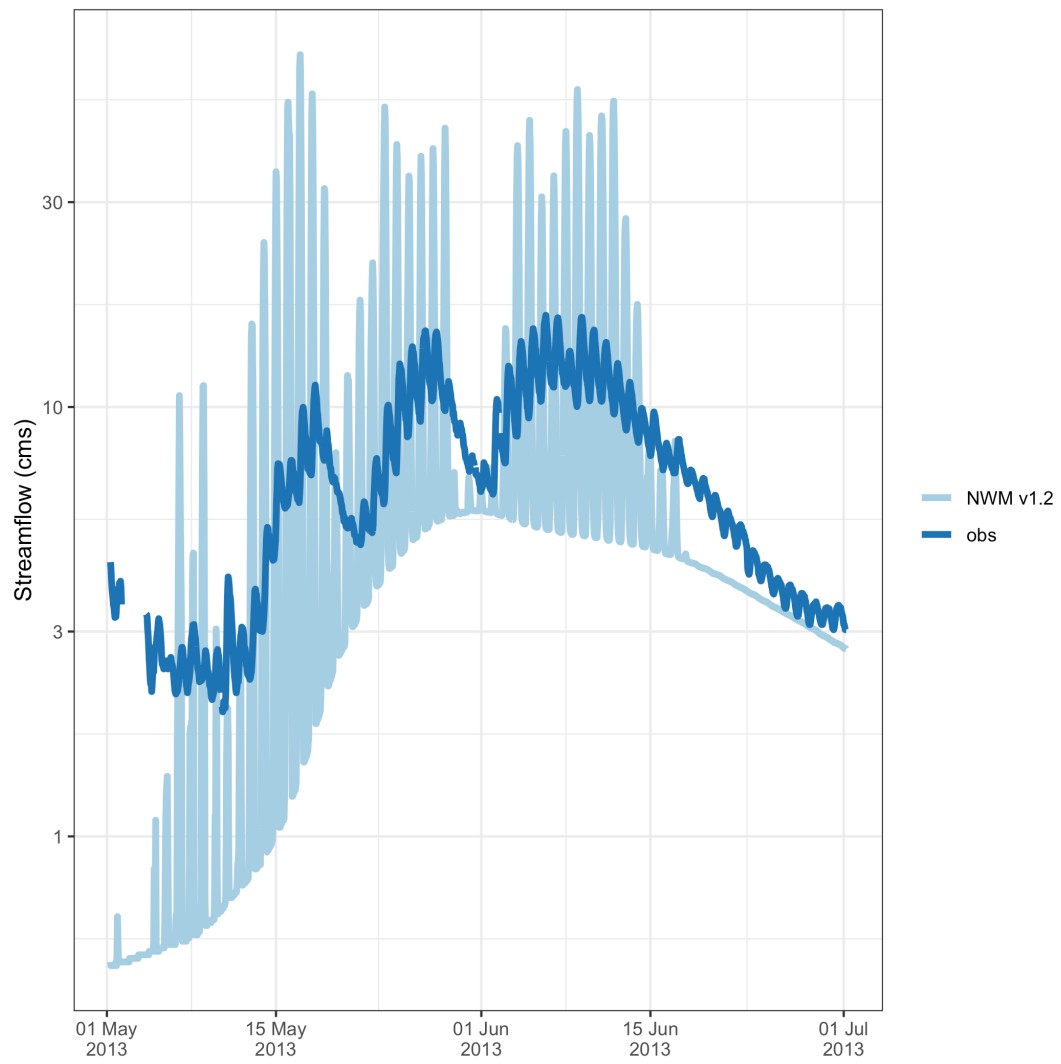

Figure 8. Taylor Park, CO: observed and simulated NWM time series.

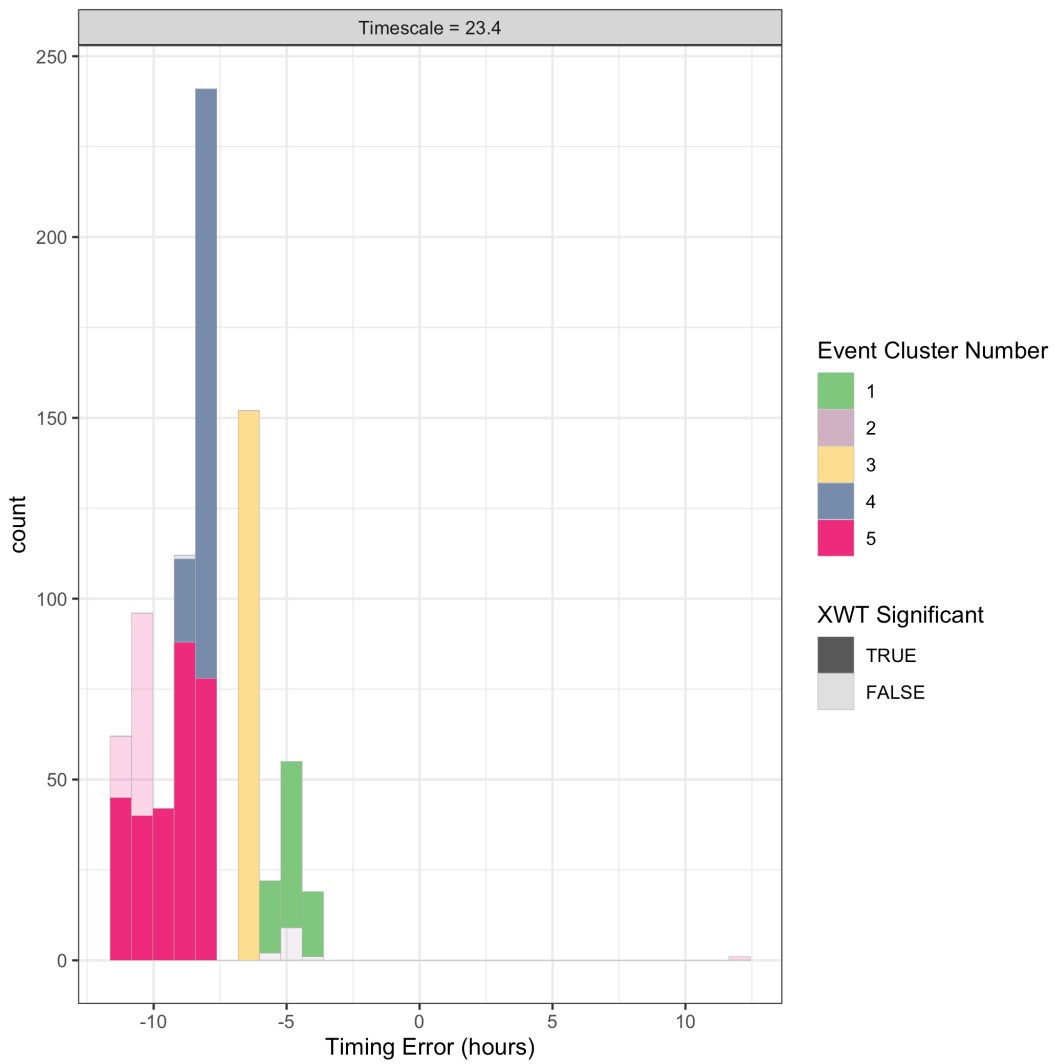

Figure 9. Taylor Park, CO: Timing error distributions for event clusters. Dark colors show if the
event was significant in the cross wavelet transform (XWT), muted colors indicate no
significance.

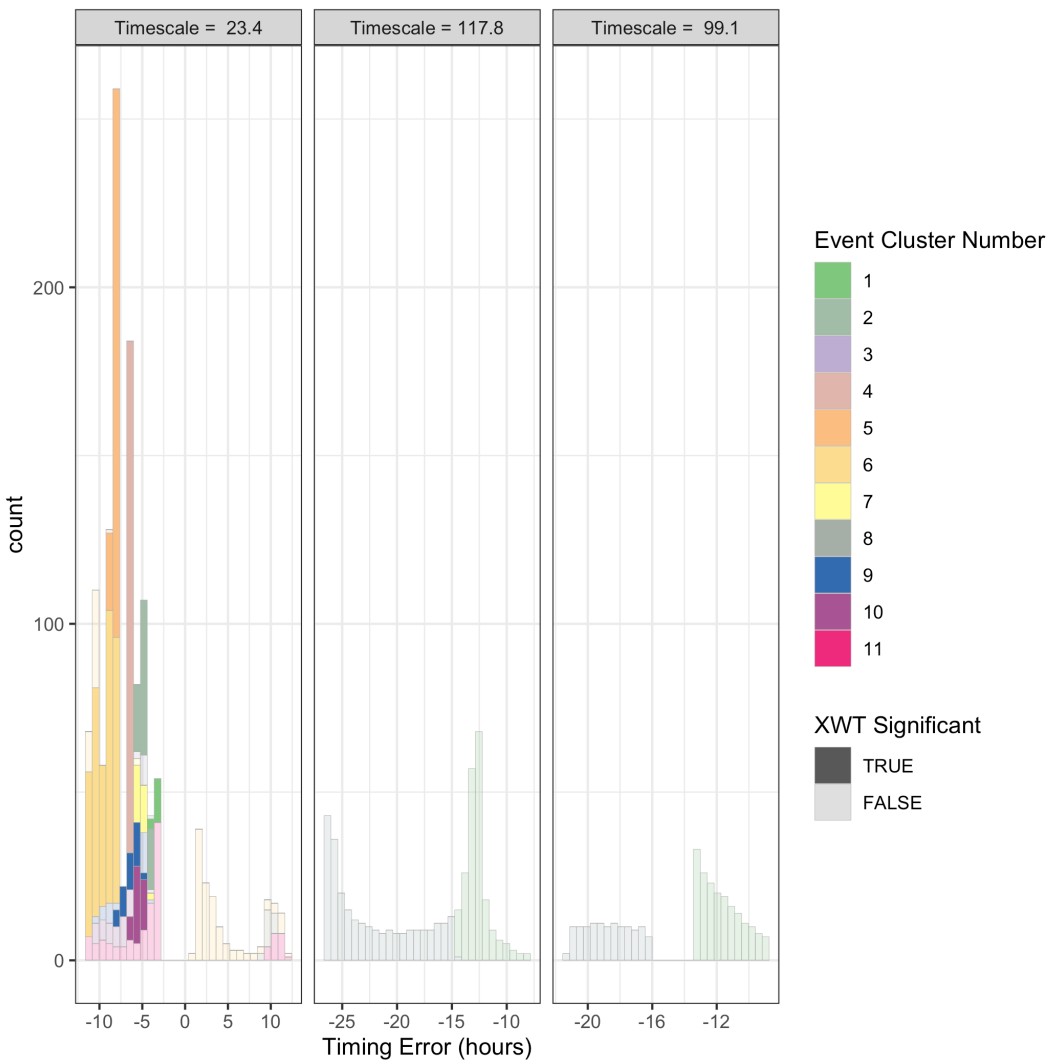

Figure 10. Taylor Park, CO: Timing error distributions for event clusters for top three
characteristic timescales (see panel title). Dark colors show if the event was significant in the
cross wavelet transform (XWT), muted colors indicate no significance.

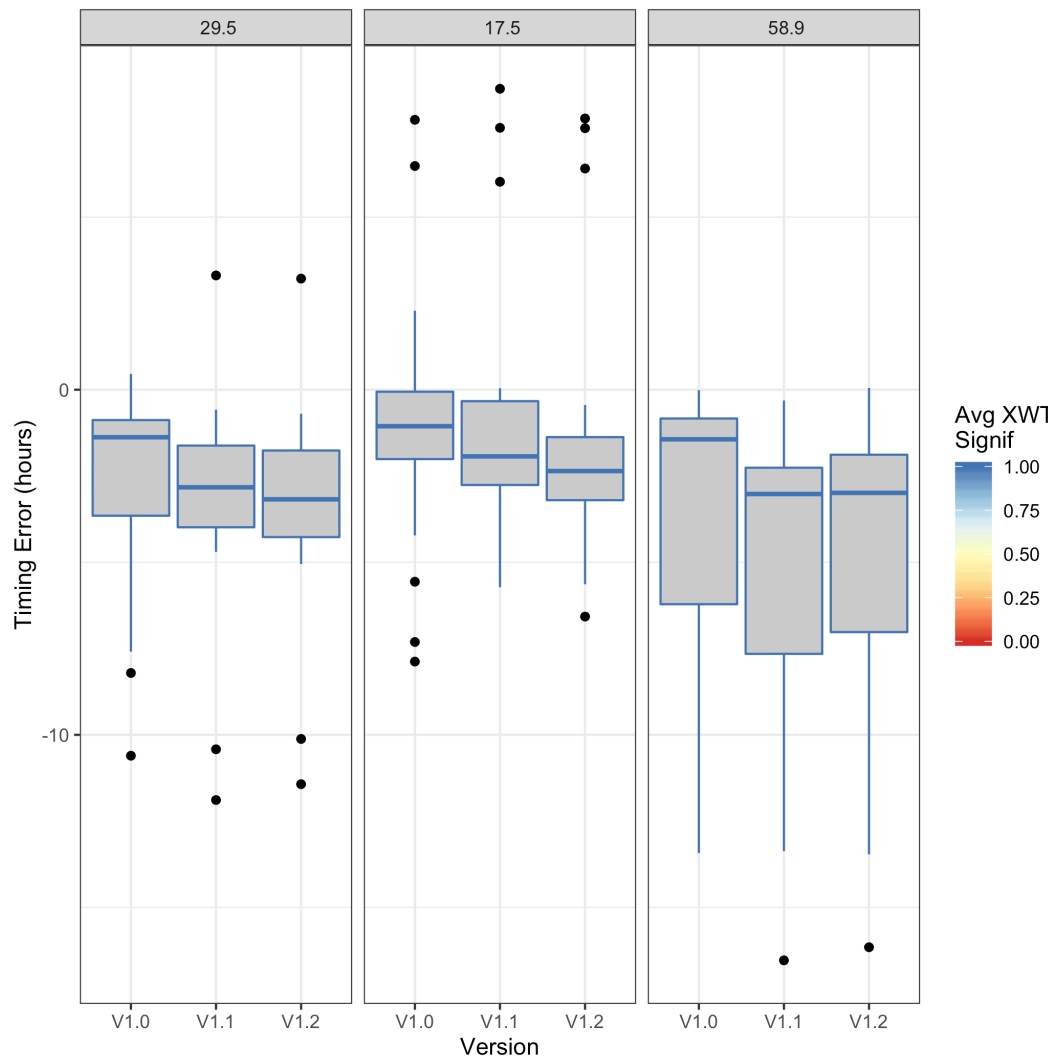

Figure 11. Five year run from Onion Creek, TX: Comparing cluster max timing error
distributions for top three characteristic timescales (see panel title) across NWM versions;
outline shading shows average significance in the cross wavelet transform (XWT).





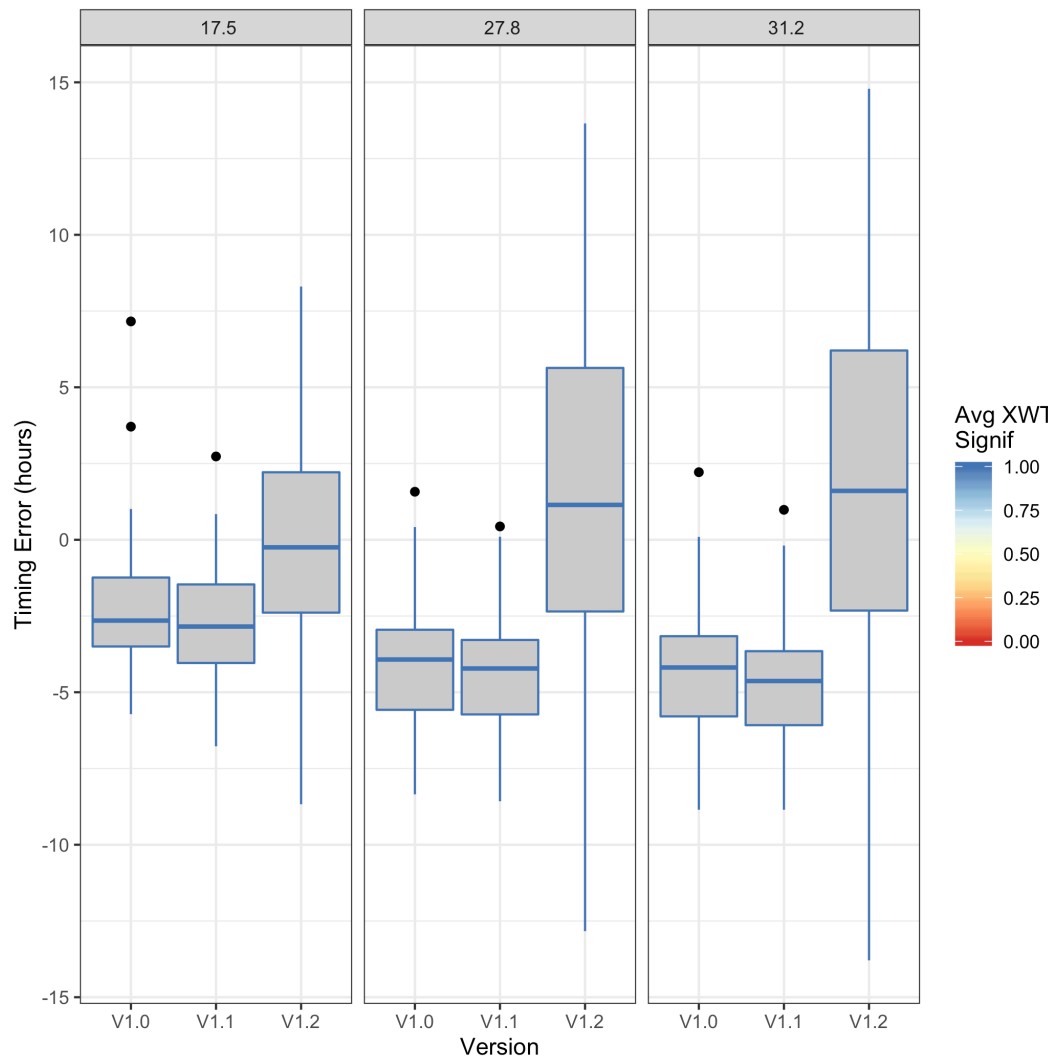

Figure 12. Five year run from Pemigewasset River, NH: Comparing cluster max timing error distributions for top three characteristic timescales (see panel title) across NWM versions; outline shading shows average significance in the cross wavelet transform (XWT).