# Peer review of "A Wavelet-Based Approach to Streamflow Event Identification and Modeled Timing Error"

_Hydrology and Earth System Sciences, 2020_

## Referee Comment (RC1) · Uwe Ehret (Referee) · 20 Oct 2020

Review of Manuscript

'A Wavelet-Based Approach to Streamflow Event Identification and Modeled Timing Error Evaluation' (hess-2020-323)

by E. Towler and J. L. McCreight

Dear Editor, dear Authors,

I have reviewed the aforementioned work. My conclusions and comments are as follows:

**1. Scope**

The article is within the scope of HESS.

**2. Summary**

The authors explain a method based on wavelet transform and cross wavelet transform to i) detect relevant events in streamflow time series and ii) to calculate timing errors between a reference time series for which the events were determined (typically an observed time series) and a test time series (typically a model simulation). Relevant regions in the full 2-d space (time and timescale) of wavelet transforms are identified by significance testing as suggested by Torrence and Compo (1998). The timing errors are calculated based on the cross wavelet transform as suggested by Liu et al. (2011), but restricted to the areas of significant events in the reference time series, which imposes a direction on the comparison. The authors illustrate their method with several application examples, which vary by their event characteristics (single event, multiple events, events caused by different processes). The authors conclude that the proposed method offers a systematic, objective, and data-driven method for event identification and timing error calculation, which can be applied to large data sets, and they stress that beyond the particular application (including particular user choices) presented in the paper, other uses of the core method are possible.

**3. Overall ranking**

Overall, the authors provide an elegant and general solution to the fuzzy problem of event detection and timing error calculation in streamflow time series, which I am sure provides more generality, better reproducibility and better insight than most existing methods, including the ones I was involved with (Series Distance). There are, however, some flaws in the study, in terms of presentation clarity and in terms of demonstrating the generality of the method beyond the particular chosen use case, which should be eliminated. The relevance of the method deserves this extra effort.

**4. Evaluation**

General points
- In the introduction, please provide a more comprehensive literature review on methods for event detection and timing error calculation. E.g. Mei and Anagnostou (2015), Merz et al. (2006), Koskelo et al. (2012).
- The use case presented in the paper takes observed events as the reference, and calculates timing errors for these (see e.g. P4 L90-92). This neglects other important aspects of event-specific (dis-)agreement of observations and simulations: False alarms and missed events. This is mentioned several times by the authors (e.g. P7 L148-153, P12 L266, P12 L271-272, P17 L388-391), and they also mention that the method could be set up differently if these aspects are of interest, but they do not explain how. False alarms and missed events are important and oftenused features of categorical model evaluation (and the idea of 'event' is categorical). So I suggest that the authors add a short discussion about if and how their method can be used to measure them (I am not asking to actually perform these analyses, but to provide guidance for future uses).

- I found it hard to follow the description of the method, as it extends over several sections of the paper:
  - In section 2, a conceptual overview is given but it misses an at least brief description of how WT and XWT function, which will be helpful for anyone not familiar with the concept. Also, section 2 refers to supplement Table 1 and supplement Fig. 1, which are in fact important to understand the method. I suggest moving these to the main paper. Section 2 refers to Fig. 1, but the concept of event clusters is not explained. This concept only becomes clear in Fig. 3, which is referred to in section 5 for the first time.
  - In section 3, the steps of the method are repeated in more detail, which creates some redundancy with section 2, but still the concept of event clusters only becomes clear in section 5 (Fig. 3). Also, I did not fully understand from the text how the observed WT and the XWT of the observed and simulated time series are related, such that evaluating the XWT in significance regions of the WT is justified (see P 12 L262-263). The authors only mention that significance areas in the WT and the XWT do not necessarily coincide (P10 L237). Please explain and justify in more detail.
  - Section 4.2 provides an application, which is somewhere in the middle between a demonstration case to explain the method (then it would be better to include it into section 3) and a demonstration cases like those in section 5. In section 4 it appears rather orphaned.
  - Overall, I suggest merging sections 2, 3 and 4.2 into one section explaining the method, which includes supplement Table 1, supplement Figure 1, and an illustrative example as shown in Fig. 3 where the concept of event clusters becomes clear.
- The meaning of 'event cluster' is not completely clear to me: From the supplement Table 1, it refers – for a particular choice of timescale – to a time-contiguous set of events (i.e. each horizontal line in Fig. 3d is one event cluster). From P9 L198-200, it seems that it refers to contiguous regions of statistical significance (i.e. the entire colored area in Fig. 3c is one event cluster). Please clarify.
- The concept of identifying relevant timescales by calculating – for every timescale - average power across all relevant events over time and then selecting local and global maxima (see e.g. Fig. 3c, right panel) is not clear to me. What is the meaning/interpretation of such a local maximum of averaged power, and how does it qualify as a selection criterion for relevant timescales? It will work when the relevant timescales are the same for all the rainfall-runoff events in the time series, but it will not if they come from different processes (such as the time series you show in supplemental Fig. 4) and have different characteristic timescales. Would it be better to assign a single characteristic timescale to each 'island of significance' (contiguous region of significant events surrounded by non-significant areas) by finding the maximum power peak in each island, and then calculating the timing error only for this representative (time,timescale)?
- I found it difficult to follow the presentation of the results in section 5, as not the same set of Figures was provided for each case in the paper. I suggest reducing the number of cases, but providing the same set of figures for all of them.

Specific points
- P3 L66: Seibert et al. (2016)
- P5 L 101: selected
- At the beginning of section 4, please add a short justification of your choice of test data
- Fig. 1d: The position and length of the horizontal green line is not clear at this point. Please explain in the text for easier comprehension.

- P14 L318: Despite what the authors state in the text, Fig. 1a and Fig. 2a do not show the same observed time series (however the significance areas in Fig. 1c and Fig. 2c are the same).
- In Fig. 2a, the 'obs' time series is light blue, in all other Figures it is dark blue. Please harmonize.
- P14 L329: I could not find Table 1
- P16 L368: I could not find Table 2
- P18 L423: I could not find Table 3
- All Figures with time series: x-axis (time) is usually given in calendar date, and y-axis (timescale) in hours. Displaying both in unit hours would facilitate the comparison of relevant timescales with the features in the time series.

Yours sincerely,

Uwe Ehret

**References**

Koskelo, A. I., Fisher, T. R., Utz, R. M., and Jordan, T. E.: A new precipitation-based method of baseflow separation and event identification for small watersheds (< 50 km2), J. Hydrol., 450–451, 267–278, https://doi.org/10.1016/j.jhydrol.2012.04.055, 2012.

Liu, Y., Brown, J., Demargne, J., Seo, D. J., 2011. A wavelet-based approach to assessing timing errors in hydrologic predictions. J. Hydrol. 397(3–4), 210–224. http://doi.org/10.1016/j.jhydrol.2010.11.040

Mei, Y. and Anagnostou, E. N.: A hydrograph separation method based on information from rainfall and runoff records, J. Hydrol., 523, 636–649, https://doi.org/10.1016/j.jhydrol.2015.01.083, 2015.

Merz, R., Blöschl, G., and Parajka, J.: Spatio-temporal variability of event runoff coefficients, J. ydrol., 331, 591–604, https://doi.org/10.1016/j.jhydrol.2006.06.008, 2006.

Torrence, C., Compo, G.P., 1998. A Practical Guide to Wavelet Analysis. Bull. Am. Meteorol. Soc. 79(1), 61-78.

---

## Short Comment (SC1) · 21 Oct 2020

Dear Uwe Ehret,

Thank you very much for your review of our paper. We look forward to addressing your feedback in detail in the near future. In our preliminary scan of your comments, we noticed that the tables were missing from the manuscript just as you noted. We included these with our submission, but somehow they were missing from what is currently available on line. We have been in touch with the journal to correct this problem. I am also going to upload those tables in the main comments area of the discussion so that they will be available to all subsequent reviewers. Sorry for this issue and thanks

for raising it.

James & Erin

---

## Short Comment (SC2) · 21 Oct 2020

Dear Editor and Reviewers,

As the initial review highlighted, the tables somehow went missing from the manuscript under review. While we have contacted the journal directly to try to fix this issue, we do not want to delay getting these tables to the reviewers. We are attaching them in this comment. Thanks for understanding and for your time.

Cheers, James & Erin

[Figure]

Please also note the supplement to this comment:
https://hess.copernicus.org/preprints/hess-2020-323/hess-2020-323-SC2-supplement.pdf

**Supplement:**

**Tables**

Table 1. Summary of timing error results for isolated peak and simulated NWM data from Onion Creek, TX, for cluster mean and max.

| Timescale (hr) | Cluster Number | Avg WT Power | Cluster Mean | | Cluster Max | |
|---|---|---|---|---|---|---|
| | | | Timing Error (hr) | % Significant in XWT | Timing Error (hr) | Significant in XWT? |
| 22 | 1 | 597743 | -3.2 | 100% | -3.2 | Yes |

Table 2. Summary of timing error results for five year run from Pemigewasset River, NH, averaged over cluster mean and max by characteristic timescale.

| Characteristic Timescale (hr) | Number of Clusters | Avg WT Power | Cluster Mean | | Cluster Max | |
|---|---|---|---|---|---|---|
| | | | Timing Error (hr) | Avg % Significance in XWT | Timing Error (hr) | Avg % Significance in XWT |
| 17.5 | 86 | 139,014 | -0.430 | 75% | -0.587 | 85% |
| 29.5 | 78 | 138,714 | 0.671 | 82% | 0.694 | 94% |
| 44.1 | 68 | 130,381 | 3.86 | 78% | 4.22 | 84% |
| 198 | 14 | 113,123 | 13.6 | 76% | 13.2 | 79% |
| 132 | 33 | 96,503 | 12.0 | 92% | 11.7 | 94% |

Table 3. Summary of timing errors from cluster max for 5-years from Onion Creek, TX.

| NWM Version | Characteristic Timescale (hr) | Number of Clusters | Avg WT Power | Median Timing Error (hr) | Avg % Significance in XWT |
|---|---|---|---|---|---|
| v1.0 | 29.5 | 17 | 4385537 | -1.38 | 100% |
| v1.1 | 29.5 | 17 | 4385537 | -2.83 | 100% |
| v1.2 | 29.5 | 17 | 4385537 | -3.18 | 100% |
| v1.0 | 17.5 | 23 | 4016123 | -1.93 | 100% |
| v1.1 | 17.5 | 24 | 3848789 | -1.06 | 100% |
| v1.2 | 17.5 | 24 | 3848789 | -2.36 | 100% |
| v1.0 | 58.9 | 11 | 2591110 | -1.44 | 100% |
| v1.1 | 58.9 | 11 | 2591110 | -3.02 | 100% |
| v1.2 | 58.9 | 11 | 2591110 | -2.99 | 100% |

Table 4. Summary of timing errors from cluster maxs for 5-years from Pemigewasset River, NH.

| NWM Version | Characteristic Timescale (hr) | Number of Clusters | Avg WT Power | Median Timing Error (hr) | Avg % Significance in XWT |
|---|---|---|---|---|---|
| v1.0 | 23.4 | 64 | 294005 | -3.42 | 84% |
| v1.1 | 23.4 | 64 | 294005 | -3.85 | 98% |
| v1.2 | 23.4 | 64 | 294005 | 1.15 | 91% |
| v1.0 | 27.8 | 61 | 284786 | -4.19 | 82% |
| v1.1 | 27.8 | 61 | 284786 | -4.21 | 97% |
| v1.2 | 27.8 | 61 | 284786 | 1.28 | 90% |
| v1.0 | 17.5 | 67 | 283064 | -2.64 | 84% |
| v1.1 | 17.5 | 67 | 283064 | -2.77 | 91% |
| v1.2 | 17.5 | 67 | 283064 | -0.51 | 85% |

---

## Referee Comment (RC2) · Cedric David (Referee) · 24 Oct 2020

General comments

The following is a review of manuscript hess-2020-323 entitled "A Wavelet-Based Approach to Streamflow Event Identification and Modeled Timing Error Evaluation" by E. Towler and J. L. McCreight being considered by Hydrology and Earth System Sciences.

This manuscripts describes a methodology for evaluating the timing of simulated river discharge hydrographs when compared with in situ observations. The approach first uses wavelet transforms (WTs) to expand observed one-dimensional hydrographs (dis-

charge vs time) into two-dimensional WTs (power vs timescale and power vs time) as a means for event detection. From those events detected in the observations, the methodology then uses cross wavelet transforms (XWTs) to evaluate the difference in timing and duration of events (at multiple power levels) between observations and simulations. The new approach is specifically designed to compare subsequent versions of a given river model and can be used both for evaluation and for diagnosis. The paper uses simulations from subsequent versions of NOAA's National Water Model and observations from the USGS at four selected locations.

I really enjoyed reading this paper and I learnt a lot from it. I agree that the evaluation of hydrograph timing is an important aspect of river model calibration/validation, and one that is often overlooked. I am guilty of that myself! This subject matter is timely given the ongoing explosion of continental scale river models such as the NWM or similar global applications. In my opinion, the authors make a strong case for the value of their work given the complexity of hydrograph shapes and describe clear guidelines for the implementation of their methodology, while also acknowledging the multiple ways in which it could be adapted. This research ought to be of interest to the readership of HESS. I elected to write this review without reading any of the other community comments so that my opinion could be relatively unbiased. My recommendation is to return the manuscript to the authors for minor revisions. My comments are outlined below, in decreasing order of importance. I hope that the authors will find some value in my suggestions. Thank you for the opportunity to review this work.

Specific comments

First, I really want to highlight that I think the authors did a commendable effort in the clarity of their explanations. I specifically enjoyed the inclusion of a "Conceptual Overview" (Section 2) and the use of the simple prescribed "+5 hours" example (Section 4.2). I think the general descriptive approach used really serves the manuscript very well and makes it accessible to many readers, including those (such as myself) who are not familiar with wavelet transform-based event detection methods. I'd like to

make two suggestions that may further help in this manner. First, I understand from Supplemental Table 1 that the units of "power" are mˆ6/sˆ2, hence they are squared units compared to discharge. It may be valuable to some readers to specify these units in the manuscript and also to perhaps suggest a hydrologic meaning to this quantity. For example, if one was to take the square root of the power value, would this in any way represent the amplitude of the peaks in the figures? If so, this may be a valuable explanation to add, which could also be graphically illustrated in Figure 1a. A rough estimate from Figure 1 suggests that the maximum power is 60,000 mˆ6/sˆ2 leading to a square root of approximately 245 mˆ3/s which is strikingly close to the amplitude of the hydrograph. Second, I could have used some further hand-holding in understanding Figure 2c. I guess I expected the entirety of the color scale to correspond to "+5 hours" (i.e. all red). I don't really understand why there appears to be a 10-hour minimum to accurately catching errors and why this makes sense because "the [time]scale must be at least double the error". It would be valuable to expand on this concept around Lines 322-325.

I wonder if there could be a good graphical way to explain how the timing error is computed in Figure 2c. The equations provided in Section 3.2.2 are not exactly straightforward, and I assume that making a graphic from those would be challenging, but it seems to be a key component of the study and some readers might benefit from such addition.

The manuscript tends to rely a bit heavily on supplemental figures throughout the text, which makes the reader jump from document to document. I suggest that the authors go through their figures carefully and evaluate whether some of supplemental figures could be combined with main manuscript figures.

The authors make a clear argument that some of the traditional error metrics (e.g. RMSE, NSE) implicitly include errors in timing but don't shed much light on them. Yet, the strength of these metrics is also the simplicity of their computation. In an effort to increase the broad acceptance and use of timing metrics that are less subjective

than peak-over-threshold, it might be helpful to for the authors to attempt a recommendation for the simplest possible form of their methodology. I understand that different powers are related to independent peaks and that there is value in looking at them all. I wonder if using powers computed as the square of discharge values from traditional occurrence probability thresholds could help reconcile (connect?) the WT approach and the threshold approach. I am not suggesting more analysis here, more so a paragraph in the discussion (Section 6, around Lines 478-487) where the authors might further expand on the simplest way for others to apply their approach.

I'm not sure I fully understand the True/False (dark grey/light grey) legend in figures 5, 9, and 10. Figure 10 seems to suggest that lighter colors are not statistically significant but it's not clear from the legend. Likewise, the "Avg XWT Signif" color bar in Figures 11 and 12 is a bit mysterious to me as I see no associated colors on the graph. Could the authors rework their legends in these figures?

Technical corrections

Line 296-297: missing "Land" in the acronym for NLDAS.

---

## Author Comment (AC1) · 16 Dec 2020

**Response to Reviewer 1**

**General Response**: We thank Reviewer 1 and Reviewer 2 for their detailed reviews, and for sharing their constructive and insightful comments. Before our point-by-point response to Reviewer 1, we note several major changes in the manuscript organization and content based on comments from both reviewers. We have addressed all review comments in our responses as well as in a substantial revision to the manuscript. The resulting manuscript is clearer and much improved. We note that for us to properly address and understand the reviews, it was necessary to actively revise the manuscript. Although the HESS process does not allow us to share our revised manuscript at this stage of the review process, we provide excerpts throughout the response to help illustrate the changes, and will provide the revised manuscript when invited.

In terms of manuscript organization, we have consolidated the methodology to be one section. Section 3 now combines previous sections 2, 3, and 4.2. This improves the clarity of the method and reduces redundancies. In terms of content, we have carefully re-evaluated what use cases, figures, and tables to include in the paper. For example, we have removed one of the locations, Bad River, SD, from the Results entirely (Section 4). In doing so, we have culled our figures and tables such that all of the figures and tables are in the main manuscript. We currently have the same number of figures as the last draft (12 Figures), but we no longer have any figures or tables in a Supplemental document. This streamlining has helped to simplify the paper and to improve its clarity. One notable example of this is that we now focus the methodology and results on cluster maxs, whereas in the previous version we included methods/results from cluster maxs and cluster means. By focusing on the cluster max analysis, we clarify how the (dis)agreement between the events in the observed WT and modeled XWT is be used to quantify "hits/misses". In summary, we thank the reviewers for bringing these points to our attention. As a result of their careful reviews, we have added numerous clarifications to the text and figures in an effort to improve overall understanding and interpretation. We hope that the editor and reviewers find these changes helpful and we look forward to sharing the revised manuscript.

Below, we provide a point-by-point response, where we address each of Reviewer 1's concerns.

**Reviewer 1.**
Dear Editor, dear Authors, Please find my comments in the attachment. Yours sincerely, Uwe Ehret

**Review of Manuscript**

**'A Wavelet-Based Approach to Streamflow Event Identification and Modeled Timing Error Evaluation' (hess-2020-323)**
**by E. Towler and J. L. McCreight**
Dear Editor, dear Authors,
I have reviewed the aforementioned work. My conclusions and comments are as follows:

**1. Scope**
The article is within the scope of HESS.

**2. Summary**
The authors explain a method based on wavelet transform and cross wavelet transform to i) detect relevant events in streamflow time series and ii) to calculate timing errors between a reference time series for which the events were determined (typically an observed time series) and a test time series (typically a model simulation). Relevant regions in the full 2-d space (time and timescale) of wavelet transforms are identified by significance testing as suggested by Torrence and Compo (1998). The timing errors are calculated based on the cross wavelet transform as suggested by Liu et al. (2011), but restricted to the areas of significant events in the reference time series, which imposes a direction on the comparison. The authors illustrate their method with several application examples, which vary by their event characteristics (single event, multiple events, events caused by different processes). The authors conclude that the proposed method offers a systematic, objective, and data-driven method for event identification and timing error calculation, which can be applied to large data sets, and they stress that beyond the particular application (including particular user choices) presented in the paper, other uses of the core method are possible.

**3. Overall ranking**
Overall, the authors provide an elegant and general solution to the fuzzy problem of event detection and timing error calculation in streamflow time series, which I am sure provides more generality, better reproducibility and better insight than most existing methods, including the ones I was involved with (Series Distance). There are, however, some flaws in the study, in terms of presentation clarity and in terms of demonstrating the generality of the method beyond the particular chosen use case, which should be eliminated. The relevance of the method deserves this extra effort.

**4. Evaluation**

General points
In the introduction, please provide a more comprehensive literature review on methods for event detection and timing error calculation. E.g. Mei and Anagnostou (2015), Merz et al. (2006), Koskelo et al. (2012).

*Thank you for this suggestion and these citations. We have added these references to expand our introduction to event detection to include baseflow separation methods. The new excerpt from the Introduction is provided here (changes in **bold**):*

*The fundamental challenge with evaluating timing errors is identifying what constitutes as an "event" in the two time series being compared. Identifying events is typically subjective, time consuming, and not practical for large-sample hydrological applications (Gupta et al. 2014).* **A variety of baseflow separation methods, ranging from physically-based to empirical, have been developed to identify hydrologic events (see Mei and Anagnostou 2015 for a summary), though many of these approaches require some manual inspection of the hydrographs. Merz et al. (2006) put forth an automated approach, but it requires a calibrated hydrologic model, which is a limitation in data poor regions. Koskelo et al. (2012) developed a simple, empirical approach that only requires rainfall and runoff time series, but is limited to small watersheds and daily data. Mei and Anagnostou (2015) introduce an automated physically-based approach, which is demonstrated for hourly data, though one caveat is that basin events need to have a clearly detectable recession period.** *Additional methods for identifying events have focused on identifying flooding events...*

The use case presented in the paper takes observed events as the reference, and calculates timing errors for these (see e.g. P4 L90-92). This neglects other important aspects of event-specific (dis-)agreement of observations and simulations: False alarms and missed events. This is mentioned several times by the authors (e.g. P7 L148-153, P12 L266, P12 L271-272, P17 L388-391), and they also mention that the method could be set up differently if these aspects are of interest, but they do not explain how. False alarms and missed events are important and often-used features of categorical model evaluation (and the idea of 'event' is categorical). So I suggest that the authors add a short discussion about if and how their method can be used to measure them (I am not asking to actually perform these analyses, but to provide guidance for future uses).

We agree that we need to enhance our description of the event (dis-)agreement of observations and simulations. We address this, but first we want to remind the reviewer of the context that, in our re-organization and streamlining, we simplified our methodology and results by focusing on cluster maxs. (In the previous version we presented results for both cluster max and cluster means approaches. Note that we also clarify the definition of cluster, please see later responses.) The cluster max is a single point of maximum power per cluster, and so cluster maxs can be classified as either hits or misses. In the previous draft we calculated the cluster mean timing error and the corresponding percent of hits within the cluster. We used the % hits in that case as a confidence measure for the timing error of each cluster. We see now how this was not clear as the hit diagnostic was different for these approaches and hence our decision to now focus the manuscript on cluster maxs. Focusing on cluster maxs simplifies our Step 2d, which was previously called "Quantify the confidence in the timing error", which we now call "Quantify Percent Hits". Percent hits now refers to a full-timeseries diagnostic of cluster maxs (instead of individual clusters). The edited section is included below:

*3.2.4. Step 2d. Quantify Percent Hits*

*The premise of computing a timing error between the observed and modeled time series is that they share common events which can be meaningfully compared. In a two-way contingency analysis of events, a "hit" refers to when the modeled timeseries reproduces an observed event. When the modeled timeseries fails to reproduce an*

*observed event, it is termed a "miss". In the case of a miss, it does not make sense to include the timing error in the overall assessment. Because the timing errors are calculated from the XWT, we choose to diagnose hits and misses based on the significance of the XWT. Once cluster maxima are selected based on the characteristic timescales of the observed event spectrum and timing errors are obtained at these locations in the XWT, the significance of the XWT on the cluster maxima is used to decide if the model produced a hit or a miss for each point. For a single cluster max, such as shown in Figure 3c, the XWT significance is either True or False, the point is either a hit or a miss. Table 2 also displays the results of the timing error analysis for this synthetic example. We can see the prescribed 5 hour offset and that the cluster maximum was significant in the XWT. When calculating timing errors for a longer time series, a useful diagnostic is to calculate the percent hits over all the cluster maxima in a timescale. When summarizing timing errors statistics for a timescale, we drop misses from the calculation and the % hits indicates what portion of the timeseries was dropped (% misses = 100 - % hits). In our tables we provided timing error statistics this way as well as over all observed events to reveal the impact of dropping misses.*

*Because, in step 1, we constrain our analysis to observed events in the wavelet power spectrum, we do not consider either of the remaining categories in a 2-way analysis (false alarms and correct negatives). We note that a complete 2-way event analysis could be constructed in the wavelet domain based on the Venn diagram of the observed and modeled events without necessarily using the XWT.*

The new Table 2 and Figure 3, referenced above, follow:

Table 2. Summary of timing error results for isolated peak and prescribed 5 hour offset from Onion Creek, TX, for cluster max.

| Characteristic Timescale (hr) | Avg WT Power | Number of Clusters | Cluster Max | | |
|---|---|---|---|---|---|
| | | | Timing Error (hr) | Time (hr) | Hit? |
| 22 | 598,000 | 1 | 5 | 37 | Yes |

[Figure]

*Figure 3. An isolated peak from Onion Creek, TX and a synthetic +5 hour offset: (a) observed and synthetic time series (note logged y-axis), (b) cross wavelet (XWT) power spectrum and phase angles (arrows), (c) sampled timing errors for observed events (dashed contour is XWT significant events) and star is cluster maximum.*

Related to the decision to focus on cluster maxs and their diagnosis as hits or misses using the XWT, earlier in the manuscript (Step 2a) we have also edited the cross wavelet (XWT) figure panels to show this visually. This is seen in Figure 3 (previously Figure2, Onion Creek synthetic example) above and in its caption. Specifically, we have adjusted panel c, to show how the observed events (colors) don't exactly overlap with the XWT significant events (dashed contour). This is now explained in the methodology:

> Similar to Step 1b of the WT, we can also calculate areas of significance for the XWT power as shown by the black contour in Figure 3b. For the XWT, significance is calculated with respect to the theoretical background wavelet spectra of each timeseries (Torrence and Compo, 1998). We define XWT events as points of significant XWT power outside the COI. XWT events indicate significant joint variability between the observed

*and modeled timeseries. In the next section, we employ XWT events as a basis for identifying hits and misses on observed events for which the timing errors are calculated. Described further in step 2d, timing errors are valid only for hits. Figure 3c shows the intersection of the observed events (colors) and the XWT events (dashed contour). This is a region of hits. Note that the early part of the observed events, particularly at shorter timescales, is not in the XWT events. This is because of the timing offset in the modeled timeseries, which misses the early part of the observed event.*

For Figures 11 and 12 (now 10 and 11) we have renamed the color scale scale "Percent Hits" and clarified the caption to include: "... outline shading shows percent (%) hits in the cross wavelet transform (XWT)." This is shown for Figure 11, below:

[Figure]

*Figure 11. Five year run from Pemigewasset River, NH: Comparing cluster max timing error distributions for top three characteristic timescales (see panel title) across NWM versions; outline shading shows percent (%) hits in the cross wavelet transform (XWT).*

In our original tables, timing errors were calculated over all observed events (i.e., both hits and misses) and we reported the percent hits (previously called "Avg % Significant in XWT"). We will calculate timing errors over only the hits for each timescale and add this as an additional column to the tables. We may drop the original timing error statistic over all observed events (hits and misses), but for now we plan to compare the two.

I found it hard to follow the description of the method, as it extends over several sections of the paper:
In section 2, a conceptual overview is given but it misses an at least brief description of how WT and XWT function, which will be helpful for anyone not familiar with the concept.

As mentioned in our general response, we have both restructured and consolidated the presentation, which allows our description of the methodology to progress logically, reduce redundancies, and allows us to clarify points of the method, such as the one the reviewer brings up here about adding additional interpretation of the wavelet transforms and associated quantities (WT, WT power, XWT, XWT power, phase, and significance (this is also partially in response to similar and additional comments from Reviewer 2). To this end, we have substantially bolstered the description of the methodology and its details in multiple places. We have inserted the following text in the methodology section to improve the paper:

> *We make several additional notes on the wavelet power and its representation in the figures. The units of the wavelet power are those of the timeseries variance (m6/s2 for streamflow) and it is natural to want to cast the power in a physical light or relate it to the timeseries variance. Indeed, the power is often normalized by the timeseries variance when presented graphically. However, it must be noted that the wavelet convolved with the timeseries frames the resulting power in terms of itself at a given scale. Wavelet power is a (normalized) measure of how well the wavelet and the timeseries match at a given time and scale. The power can only be compared to other values of power resulting from a similarly constructed WT. There are various transforms that can be applied to aid graphical interpretation of the power (log, variance scaling), but the utility of these often depends on the nature of the individual timeseries analyzed. For simplicity, we plot the raw bias-rectified wavelet power in this paper.*

Also, section 2 refers to supplement Table 1 and supplement Fig. 1, which are in fact important to understand the method. I suggest moving these to the main paper.

We agree, and have moved these to the main paper: Supplemental Figure 1 is now Figure 1 and Supplemental Table 1 is now Table 1.

Section 2 refers to Fig. 1, but the concept of event clusters is not explained. This concept only becomes clear in Fig. 3, which is referred to in section 5 for the first time.

As mentioned, per the reviewer's suggestion, we have restructured the methodology (now Section 3) by merging draft sections 2, 3, and 4.2. At the very beginning of Section 3, we introduce Table 1 (previously Supplemental Table 1), which is the nomenclature table of terms used in the manuscript, which includes the definition of "event cluster", which can be used for reference as the reader progresses through the steps of the methodology. Further, in our restructuring of Section 3, we now illustrate the steps of the methodology by using the observed time series of an isolated peak in Onion Creek, TX, (which was previously referred to and illustrated in separate sections); hence, now the concept of event clusters is defined and a figure illustrating it is referred to for the first time, at the same point in the manuscript. We

provide an excerpt to where this occurs in the methodology here, which occurs in section 3.1.3 (Step 1c), with the definition of clusters in bold:

> •*Identify timescales of absolute and local average power maxima: After obtaining the average event power as a function timescale (Figure 2c, right), the local and absolute maximums for average event power can be determined. In the Onion Creek case, there is a single maximum at 22 hours (grey dot in Figure 2c, right panel). The timescales corresponding to the absolute and local maxima of the average power of the observed time series are called the characteristic timescales used for evaluation. This is the first subset of events: all events that fall within the characteristic timescales.* **For a single characteristic timescale, contiguous events in time are called event clusters (horizontal line in Figure 2d)**.

[Figure]

Figure 2. An isolated peak from Onion Creek, TX: (a) observed time series, (b) observed wavelet power spectrum (left) and average power by timescale for all points (right); (c) statistically significant wavelet power spectrum or events (left) and average power by time scale for all events with maxima shown by grey dots (right); (d) Characteristic scale event cluster (horizontal green line) and cluster maximum (star).

In section 3, the steps of the method are repeated in more detail, which creates some redundancy with section 2, but still the concept of event clusters only becomes clear in section 5 (Fig. 3).

See previous response.

Also, I did not fully understand from the text how the observed WT and the XWT of the observed and simulated time series are related, such that evaluating the XWT in significance regions of the WT is justified (see P 12 L262-263). The authors only mention that significance areas in the WT and the XWT do not necessarily coincide (P10 L237). Please explain and justify in more detail.

We addressed this comment above where we enhanced the description of XWT significance and events in the context of modeled hits and misses.

Section 4.2 provides an application, which is somewhere in the middle between a demonstration case to explain the method (then it would be better to include it into section 3) and a demonstration cases like those in section 5. In section 4 it appears rather orphaned.

Overall, I suggest merging sections 2, 3 and 4.2 into one section explaining the method, which includes supplement Table 1, supplement Figure 1, and an illustrative example as shown in Fig. 3 where the concept of event clusters becomes clear.

We agree, and in summary, we have merged these sections, moved Supplemental Table 1 and Supplemental Figure 1 to the main manuscript, and better clarified the concept of event clusters.

The meaning of 'event cluster' is not completely clear to me: From the supplement Table 1, it refers – for a particular choice of timescale – to a time-contiguous set of events (i.e. each horizontal line in Fig. 3d is one event cluster). From P9 L198-200, it seems that it refers to contiguous regions of statistical significance (i.e. the entire colored area in Fig. 3c is one event cluster). Please clarify.

See previous response.

The concept of identifying relevant timescales by calculating – for every timescale - average power across all relevant events over time and then selecting local and global maxima (see e.g. Fig. 3c, right panel) is not clear to me. What is the meaning/interpretation of such a local maximum of averaged power, and how does it qualify as a selection criterion for relevant timescales? It will work when the relevant timescales are the same for all the rainfall-runoff events in the time series, but it will not if they come from different processes (such as the time series you show in supplemental Fig. 4) and have different characteristic timescales. Would it be better to assign a single characteristic timescale to each 'island of significance' (contiguous region of significant events surrounded by non-significant areas) by finding the maximum power peak in each island, and then calculating the timing error only for this representative (time,timescale)?

This is a good point, and made us realize two things: First, that we need to better justify our decision to average in timescale; and second that we need to acknowledge that there are other

ways to identify events for which to calculate the timing errors, such as using "islands of significance". In terms of the former, for this paper, we wanted to provide a technique that (a) builds off previous work, and (b) is as simple and straightforward as possible. In terms of the former, Torrence and Compo (1998) offer two methods to smoothing the wavelet plot that can increase significance and confidence: (i) averaging in time or (ii) averaging in timescale. In this paper, we average in timescale, since that can reveal the dominant timescales on which events are occuring, which is useful to model diagnostics. The assumption is that identifiable sets of processes of interest are distinct in timescale, and that averaging over many events will reveal its expected value. It is true that this means that sometimes, there may be a maximum power peak below or above the identified characteristic timescale, but that this is smoothed out by averaging over all the significant events. We clarify this point now in the methodology, specifically in Step 1c, the excerpt for this is included below (changes in bold):

*3.1.3. Step 1c. Sample observed events to an event-set relevant to analysis*

*Step 1b results in the identification of all events at all timescales and times. In this sub-step, the event space is sampled to suit the particular evaluation.* ***Torrence and Compo (1998) offer two methods to smoothing the wavelet plot that can increase significance and confidence: (i) averaging in time (over timescale) or (ii) averaging in timescale (over time). Because the goal of this paper is to evaluate model timing errors over long simulation periods, we choose to sample the event space based on averaging in timescale. Although for some locations there may be physical reasons to expect certain timescales to be important (e.g., seasonal cycle of snowmelt), the most important scales at which hydrologic signals occur at a particular location are not necessarily known a priori. Averaging events in timescale can provide a useful diagnostic by identifying the dominant, or "characteristic", timescales for a given time series. Averaging many events in timescale can filter noise and help reveal the expected timescales of dominant variability corresponding to different processes or sets of processes***.

In terms of the latter, using "island of significance" was one of our first ideas when we set out to quantify timing errors. However, this approach has several drawbacks: 1) selecting a single peak ignores that multiple frequencies can be important at once; this is illustrated with Figure 4c, below, (which was draft Figure 3c), where for the Pemigewasset River events shown, there are islands of significance that include events for different characteristic timescales (i.e., there are 3 characteristic time scales around 24 hours and 2 characteristic timescales at 111 and 148 hours):

[Figure]

*Figure 4. Multiple peaks from Pemigewasset River, NH: (a) observed time series, (b) observed wavelet power spectrum (left) and average power by timescale for all points (right); (c) statistically significant wavelet power spectrum or events (left) and average power by time scale for all events with maxima shown by grey dots (right); (d) Characteristic scales event clusters (horizontal lines).*

However, there is a second drawback, which is that defining islands when connected is problematic (e.g., How do we define the islands? What if there is one small connecting point connecting 2 islands somewhere in the time/timescale WT?). Averaging in timescale is a more straightforward approach for model diagnostics. However, if one thought that the characteristic timescales were non-stationary, i.e., changing over the length of the time series, then you could do moving timescale averaging (our approach with moving windows) to investigate the non-stationarity. We acknowledge this in the Discussion and Conclusions:

> *We only look at the timing errors from an event-set relevant to our analysis, but there are other ways to subset the events that might be more suitable to other applications. For instance, we focus on the cluster max, but one could also examine the cluster mean. Also, instead of finding the event of maximum power in each cluster (i.e., for a given timescale), it would be possible to identify the event with maximum power in "islands of significance", i.e., significant areas contiguous in time across timescales. However, this ignores that multiple frequencies can be important at once and defining the islands when connected is problematic. If one suspected non-stationarity in the characteristic*

*timescales over the timeseries, then a different approach such as a moving average in timescale could be employed.*

I found it difficult to follow the presentation of the results in section 5, as not the same set of Figures was provided for each case in the paper. I suggest reducing the number of cases, but providing the same set of figures for all of them.

We agree with the Reviewer, and as mentioned, we have carefully reviewed the content of the manuscript to address this. First, in our new Methodology (Section 3), we illustrate the method using the Onion Creek, TX, isolated peak observation and a prescribed timing error (not NWM simulation data). In the Results (Section 4), we further demonstrate the method, increasing the complexity by using NWM simulated data and longer time series. As suggested, we reduce the number of cases, and we remove the example of Bad River. We still look at two examples: Pemigewasset River, NH, and Taylor River, CO, but in the first subsection (section 4.1) we only look at the 3-month time series for Pemigewasset River, and a one-year time series for Taylor River, showing the XT and XWT figures for both. We also only focus on the cluster max (in the previous version we had a comparison of cluster max and cluster mean). We now acknowledge that one could also look at cluster mean in the Discussion. In the second Results subsection (4.2), we still show version-over-version comparisons for 5-year simulations for Onion Creek, Pemigewasset River, and Taylor River, to illustrate the utility for evaluation.

For the benefit of the reviewer, here is a summary table mapping the old figures to the new figures, as well as a description of the figures now included:

| NEW | OLD | Description |
| --- | --- | --- |
| Figure 1 | Supplemental Figure 1 | Flow chart of methodology |
| Figure 2 | Figure 1 | Isolated Peak Onion Creek, Step 1 (WT) |
| Figure 3 | Figure 2 | Isolated Peak Onion Creek, Step 2 (XWT), synthetic +5 hour offset |
| Figure 4 | Figure 3 | 3-month Pemigewasset River, Step 1 (WT) |
| Figure 5 | Figure 4 | 3-month Pemigewasset River, Step 2 (XWT), NWM v1.2 |
| Figure 6 | Figure 5 | 3-month Pemigewasset River, Cluster timing errors by characteristic scale |
| Figure 7 | Supplemental Figure 4 | 1-year Taylor River, Step 1 (WT) |
| Figure 8 | NA (new figure) | 1-year Taylor River, Step 2 (XWT), NWM v1.2 |
| Figure 9 | Figure 8 | 1-year Taylor River, zoom in of spring runoff obs and NWM v1.2 time series |
| Figure 10 | Figure 11 | 5-year Onion Creek, NWM version comparison, cluster max timing errors |
| Figure 11 | Figure 12 | 5-year Pemigewasset River, NWM version comparison, cluster max timing errors |
| Figure 12 | Supplemental Figure 7 | 5-year Taylor River, peak streamflows by characteristic scale |

Further, here are the figures that have been removed from the manuscript, along with the reason:

| Cut figures (OLD) | Description | Reason cut |
|---|---|---|
| Supplmental Figure 2 | Isolated Peak Onion Creek, Step 2 (XWT), NWM v1.2 | Need to reduce use cases and was repetative; got cut when we consolidated methods section in re-structuring. |
| Supplmental Fig 3 | 5-year Pemigewasset River, NWM v1.2, compared cluster max with cluster mean timing errors | Need to reduce use cases; showing mean vs max is unnecessary (now just mention in Discussion) |
| Figure 9 | Spring runoff for Taylor River, Cluster timing errors by characteristic scale | Need to reduce use cases, already see clusters once for Pemi (Fig 6) |
| Figure 10 | 1-year Taylor River, Cluster timing errors by characteristic scale | Need to reduce use cases, already see clusters once for Pemi (Fig 6); Table 4 now summarizes this. |
| Sup Fig 5 | 5-year Taylor River, NWM version comparison, cluster max timing errors. | Need to reduce figures, already show this for Onion and Pemi, Table 7 describes this sufficiently. |
| Sup Fig 6 | 1-year Bad River, NWM v1.2 | Need to reduce use cases |
| Figure 6 | 2-months Bad River, Step 1 (WT) | Need to reduce use cases |
| Figure 7 | 2-months Bad River, Step 2 (XWT), NWM v1.2 | Need to reduce use cases |

Specific points
P3 L66: Seibert et al. (2016)

Thank you, we have added the "." after al

P5 L 101: selected

This has been corrected.

At the beginning of section 4, please add a short justification of your choice of test data

These locations represent three different regions in the U.S., namely South Central, New England, and Intermountain West. We have added text to this effect in the Data section (now Section 2):

> "2. Data
>
> The application of the methodology is illustrated using real and simulated stream discharge (streamflow, m3/s) data from U.S. Geological Survey (USGS) stream gage locations representing three different geographic regions: Onion Creek at US Highway 183, Austin, Texas for the South Central region (Onion Creek, TX; USGS site number 08159000), Taylor River at Taylor Park, Colorado for the Intermountain West (Taylor River, CO; USGS site number 09107000), and Pemigewasset River at Woodstock, New Hampshire for New England (Pemigewasset River, NH; USGS site number 01075000)."

Fig. 1d: The position and length of the horizontal green line is not clear at this point. Please explain in the text for easier comprehension.
This is related to the clarification of the term event cluster, this has been addressed through the merging of the methodology sections and a clarification of the definition (see previous responses).

P14 L318: Despite what the authors state in the text, Fig. 1a and Fig. 2a do not show the same observed time series (however the significance areas in Fig. 1c and Fig. 2c are the same).

Thanks for pointing this out. We agree this can be confusing. These look different because the observed streamflow time series (Fig 1a in previous version) is in regular/linear scale and the combined observed and simulated time series on a logged scale (Fig 2a in previous version). We do this because the log scale helps reveal differences in the two time series. We now clarify this in the text when we first introduce the figure (now Figure 3): "... and the synthetic modeled time series which is uniformly shifted 5 hours to the future (figure 3a, note the log scale)" and in the caption: "Figure 3. An isolated peak from Onion Creek, TX and a synthetic +5 hour offset: (a) observed and synthetic time series (note logged y-axis)."

In Fig. 2a, the 'obs' time series is light blue, in all other Figures it is dark blue. Please harmonize.

Indeed. Thanks for your attention to detail! You can see the updated figure (the first figure, captioned Figure 3) with the fix above.

P14 L329: I could not find Table 1

We apologize that the tables were not included in the original manuscript you reviewed, and we were thankful that you raised this issue. Once we received your review, we uploaded the tables to the discussion, and they are now included in the revised manuscript.

P16 L368: I could not find Table 2

See above response.

P18 L423: I could not find Table 3

See above response.

All Figures with time series: x-axis (time) is usually given in calendar date, and y-axis (timescale) in hours. Displaying both in unit hours would facilitate the comparison of relevant timescales with the features in the time series.

Thanks for the suggestion. We are considering this for the final set of figures. We may set the minor grid resolution to hours. We see the potential value, but success will be plot dependent as the time series length can make clear presentation difficult.

Yours sincerely,
Uwe Ehret

**References**

Koskelo, A. I., Fisher, T. R., Utz, R. M., and Jordan, T. E.: A new precipitation-based method of baseflow separation and event identification for small watersheds (< 50 km2), J. Hydrol., 450–451, 267–278, https://doi.org/10.1016/j.jhydrol.2012.04.055, 2012.

Liu, Y., Brown, J., Demargne, J., Seo, D. J., 2011. A wavelet-based approach to assessing timing errors in hydrologic predictions. J. Hydrol. 397(3–4), 210–224. http://doi.org/10.1016/j.jhydrol.2010.11.040

Mei, Y. and Anagnostou, E. N.: A hydrograph separation method based on information from rainfall and runoff records, J. Hydrol., 523, 636–649, https://doi.org/10.1016/j.jhydrol.2015.01.083, 2015.

Merz, R., Blöschl, G., and Parajka, J.: Spatio-temporal variability of event runoff coefficients, J. Hydrol., 331, 591–604, https://doi.org/10.1016/j.jhydrol.2006.06.008, 2006.

Torrence, C., Compo, G.P., 1998. A Practical Guide to Wavelet Analysis. Bull. Am. Meteorol. Soc. 79(1), 61-78.

==================================================================

---

## Author Comment (AC2) · 16 Dec 2020

**Response to Reviewer 2**

**General Response**: We thank Reviewer 1 and Reviewer 2 for their detailed reviews, and for sharing their constructive and insightful comments. Before our point-by-point response to Reviewer 2, we note several major changes in the manuscript organization and content based on comments from both reviewers. We have addressed all review comments in our responses as well as in a substantial revision to the manuscript. The resulting manuscript is clearer and much improved. We note that for us to properly address and understand the reviews, it was necessary to actively revise the manuscript. Although the HESS process does not allow us to share our revised manuscript at this stage of the review process, we provide excerpts throughout the response to help illustrate the changes, and will provide the revised manuscript when invited.

In terms of manuscript organization, we have consolidated the methodology to be one section. Section 3 now combines previous sections 2, 3, and 4.2. This improves the clarity of the method and reduces redundancies. In terms of content, we have carefully re-evaluated what use cases, figures, and tables to include in the paper. For example, we have removed one of the locations, Bad River, SD, from the Results entirely (Section 4). In doing so, we have culled our figures and tables such that all of the figures and tables are in the main manuscript. We currently have the same number of figures as the last draft (12 Figures), but we no longer have any figures or tables in the Supplemental document. This streamlining has helped to simplify the paper and to improve its clarity. One notable example of this is that we now focus the methodology and results on cluster maxs, whereas in the previous version we included methods/results from cluster maxs and cluster means. By focusing on the cluster max analysis, we clarify how the (dis)agreement between the events in the observed WT and modeled XWT can be used to quantify "hits/misses". In summary, we thank the reviewers for bringing these points to our attention. As a result of their careful reviews, we have added numerous clarifications to the text and figures in an effort to improve overall understanding and interpretation. We hope that the editor and reviewers find these changes helpful and we look forward to sharing the revised manuscript.

Below, we provide a point-by-point response, where we address each of Reviewer 2's concerns.

**Reviewer 2.**
Cedric David (Referee) cedric.david@jpl.nasa.gov

General comments

The following is a review of manuscript hess-2020-323 entitled "A Wavelet-Based Approach to Streamflow Event Identification and Modeled Timing Error Evaluation" by E. Towler and J. L. McCreight being considered by Hydrology and Earth System Sciences.

This manuscripts describes a methodology for evaluating the timing of simulated river discharge hydrographs when compared with in situ observations. The approach first uses wavelet

transforms (WTs) to expand observed one-dimensional hydrographs (discharge vs time) into two-dimensional WTs (power vs timescale and power vs time) as a means for event detection. From those events detected in the observations, the methodology then uses cross wavelet transforms (XWTs) to evaluate the difference in timing and duration of events (at multiple power levels) between observations and simulations. The new approach is specifically designed to compare subsequent versions of a given river model and can be used both for evaluation and for diagnosis. The paper uses simulations from subsequent versions of NOAA's National Water Model and observations from the USGS at four selected locations.

I really enjoyed reading this paper and I learnt a lot from it. I agree that the evaluation of hydrograph timing is an important aspect of river model calibration/validation, and one that is often overlooked. I am guilty of that myself! This subject matter is timely given the ongoing explosion of continental scale river models such as the NWM or similar global applications. In my opinion, the authors make a strong case for the value of their work given the complexity of hydrograph shapes and describe clear guidelines for the implementation of their methodology, while also acknowledging the multiple ways in which it could be adapted. This research ought to be of interest to the readership of HESS. I elected to write this review without reading any of the other community comments so that my opinion could be relatively unbiased. My recommendation is to return the manuscript to the authors for minor revisions. My comments are outlined below, in decreasing order of importance. I hope that the authors will find some value in my suggestions. Thank you for the opportunity to review this work.

Thank you very much for this helpful global assessment. Your specific suggestions are extremely valuable and have helped strengthen the presentation and content. Thank you for your careful review. To refer to the different responses, we have numbered each Reviewer response.

**Specific comments**

First, I really want to highlight that I think the authors did a commendable effort in the clarity of their explanations. I specifically enjoyed the inclusion of a "Conceptual Overview" (Section 2) and the use of the simple prescribed "+5 hours" example (Section 4.2). I think the general descriptive approach used really serves the manuscript very well and makes it accessible to many readers, including those (such as myself) who are not familiar with wavelet transform-based event detection methods. I'd like to make two suggestions that may further help in this manner. First, I understand from Supplemental Table 1 that the units of "power" are $m^6/s^2$, hence they are squared units compared to discharge. It may be valuable to some readers to specify these units in the manuscript and also to perhaps suggest a hydrologic meaning to this quantity. For example, if one was to take the square root of the power value, would this in any way represent the amplitude of the peaks in the figures? If so, this may be a valuable explanation to add, which could also be graphically illustrated in Figure 1a. A rough estimate from Figure 1 suggests that the maximum power is 60,000 $m^6/s^2$ leading to a square root of approximately 245 $m^3/s$ which is strikingly close to the amplitude of the hydrograph.

R.2.1. Before we respond to this specific inquiry, we want to point out that as noted in our General Response, we have restructured the methodology (now Section 3) by merging draft sections 2, 3, and 4.2. This allows our description of the methodology to progress logically, reduce redundancies, and allows us to clarify points of the method, such as the one the reviewer brings up here.

The point brought up here is worthy of discussion generally: additional interpretation of the wavelet transforms and associated quantities (WT, WT power, XWT, XWT power, phase, and significance). Reviewer 1 also made a similar suggestion and this is a point echoed later in this review as well. We have substantially bolstered the description of the methodology and its details in multiple places. With regards to the interpretation of the wavelet power raised here, we have inserted the following text in the methodology section to improve the paper:

> *We make several additional notes on the wavelet power and its representation in the figures. The units of the wavelet power are those of the timeseries variance (m6/s2 for streamflow) and it is natural to want to cast the power in a physical light or relate it to the timeseries variance. Indeed, the power is often normalized by the timeseries variance when presented graphically. However, it must be noted that the wavelet convolved with the timeseries frames the resulting power in terms of itself at a given scale. Wavelet power is a (normalized) measure of how well the wavelet and the timeseries match at a given time and scale. The power can only be compared to other values of power resulting from a similarly constructed WT. There are various transforms that can be applied to aid graphical interpretation of the power (log, variance scaling), but the utility of these often depends on the nature of the individual timeseries analyzed. For simplicity, we plot the raw bias-rectified wavelet power in this paper.*

Second, I could have used some further hand-holding in understanding Figure 2c. I guess I expected the entirety of the color scale to correspond to "+5 hours" (i.e. all red). I don't really understand why there appears to be a 10-hour minimum to accurately catching errors and why this makes sense because "the [time]scale must be at least double the error". It would be valuable to expand on this concept around Lines 322-325.

R2.2. We fully agree that this should be better explained. In fact, the "phase aliasing" that can be seen at timescales shorter than 10 hrs (2x the synthetic timing error) was something we hoped the reviewers would ask about and that we could expand our discussion in that context. We reworked and expanded the description of the cross-wavelet phase difference with discussion in context of the synthetic example as follows in our revised manuscript:

> *3.2.2. Step 2b. Calculate the cross-wavelet timing errors*
> *For complex wavelets, such as the Morlet used in this paper, the individual WTs include an imaginary component of the convolution. Together, the real and imaginary parts of the convolution describe the phase of each timeseries with respect to the wavelet. The cross wavelet transform combines the WTs in conjugate, allowing the calculation of a phase difference or angle (radians) which can be computed as:*

$$\phi_n^{XY}(s) = tan^{-1}\left[\frac{\Im(\langle s^{-1}W_n^{XY}(s)\rangle)}{\Re(\langle s^{-1}W_n^{XY}(s)\rangle)}\right],$$

*Where I is the imaginary and R is the real component of W_n^XY (s). The arrows in Figure 3b indicate the phase difference for our example case, which are used to calculate the timing errors. Note that these are calculated at all points in the wavelet domain.*

*The distance around the phase circle at each timescale is the Fourier period (hours). We convert the phase angle into the timing errors (hours) as in Liu et al. (2011):*

$$\Delta t_n^{XY}(s) = \phi_n^{XY}(s) * T/2\pi$$

*where T is the equivalent Fourier period of the wavelet. Note that the maximum timing error which can be represented at each timescale is half the Fourier period because the phase angle is in the interval (-pi, pi). In other words, only timescales greater than 2E can accurately represent a timing error E. Because of the phase discontinuity at ±pi, true phase angles outside this range alias to angles inside this range. (For example, the phase angles 1.05 * pi and -.95 * pi are both represented by the latter). When the wavelet transforms are approximately antiphase, the computed phase differences and timing errors produce bimodal distributions due to noise in the data. Figure 3c shows phase aliasing in the negative timing errors at timescales less than 10 hours, double the 5 hour synthetic timing error we introduced. The bimodality of the phase and timing are also seen at the 10hr timescale when the timing errors abruptly change sign (or phase by 2pi). We note the convention used is that the XWT produces timing errors that are interpreted as "modeled minus observed", i.e., positive values mean the model occurs after the observed. Positive 5 hour timing errors in Figure 3c describe that the model is "late" compared to the observations as seen in the hydrographs in the top panel (a).*

I wonder if there could be a good graphical way to explain how the timing error is computed in Figure 2c. The equations provided in Section 3.2.2 are not exactly straightforward, and I assume that making a graphic from those would be challenging, but it seems to be a key component of the study and some readers might benefit from such addition.

R2.3. We see the reviewer's point, and note that in the previous draft, we had aspects describing and demonstrating methodology spread across several sections (Section 2, Section 3, and Section 4.2); whereas in this new version, we have consolidated those sections into a single section (now Section 3). We hope that by having the equations and the illustrative example (i.e., Onion Creek isolated peak) all in one place, we will help to better guide the reader. We concur that an illustration would be challenging and have elected to more comprehensively describe the complex algebra involved in the calculation of the phase angle and the timing error, as per the response to the previous comment (R2.2). We have also

bolstered the description of the XWT power and how it is used to diagnose if timing errors are "hits"; this is further described in our final response, R2.6.

The manuscript tends to rely a bit heavily on supplemental figures throughout the text, which makes the reader jump from document to document. I suggest that the authors go through their figures carefully and evaluate whether some of supplemental figures could be combined with main manuscript figures.

R2.4 We agree, and as mentioned, we have carefully reviewed the content of the manuscript to address this. As a result, we have culled our figures and tables, so that now all of the figures and tables are now in the manuscript. This required removing one of the use cases: the Bad River, SD, as well as additional streamlining. We currently have the same number of figures as the last draft (12 Figures), but we no longer have any figures or tables in the Supplemental.

For the benefit of the reviewer, here is a summary table mapping the old figures to the new figures, as well as a description of the figures now included:

| NEW | OLD | Description |
|---|---|---|
| Figure 1 | Supplemental Figure 1 | Flow chart of methodology |
| Figure 2 | Figure 1 | Isolated Peak Onion Creek, Step 1 (WT) |
| Figure 3 | Figure 2 | Isolated Peak Onion Creek, Step 2 (XWT), synthetic +5 hour offset |
| Figure 4 | Figure 3 | 3-month Pemigewasset River, Step 1 (WT) |
| Figure 5 | Figure 4 | 3-month Pemigewasset River, Step 2 (XWT), NWM v1.2 |
| Figure 6 | Figure 5 | 3-month Pemigewasset River, Cluster timing errors by characteristic scale |
| Figure 7 | Supplemental Figure 4 | 1-year Taylor River, Step 1 (WT) |
| Figure 8 | NA (new figure) | 1-year Taylor River, Step 2 (XWT), NWM v1.2 |
| Figure 9 | Figure 8 | 1-year Taylor River, zoom in of spring runoff obs and NWM v1.2 time series |
| Figure 10 | Figure 11 | 5-year Onion Creek, NWM version comparison, cluster max timing errors |
| Figure 11 | Figure 12 | 5-year Pemigewasset River, NWM version comparison, cluster max timing errors |
| Figure 12 | Supplemental Figure 7 | 5-year Taylor River, peak streamflows by characteristic scale |

Further, here are the figures that have been removed from the manuscript, along with the reason:

| Cut figures (OLD) | Description | Reason cut |
|---|---|---|
| Supplmental Figure 2 | Isolated Peak Onion Creek, Step 2 (XWT), NWM v1.2 | Need to reduce use cases and was repetative; got cut when we consolidated methods section in re-structuring. |
| Supplmental Fig 3 | 5-year Pemigewasset River, NWM v1.2, compared cluster max with cluster mean timing errors | Need to reduce use cases; showing mean vs max is unnecessary (now just mention in Discussion) |
| Figure 9 | Spring runoff for Taylor River, Cluster timing errors by characteristic scale | Need to reduce use cases, already see clusters once for Pemi (Fig 6) |
| Figure 10 | 1-year Taylor River, Cluster timing errors by characteristic scale | Need to reduce use cases, already see clusters once for Pemi (Fig 6); Table 4 now summarizes this. |
| Sup Fig 5 | 5-year Taylor River, NWM version comparison, cluster max timing errors. | Need to reduce figures, already show this for Onion and Pemi, Table 7 describes this sufficiently. |
| Sup Fig 6 | 1-year Bad River, NWM v1.2 | Need to reduce use cases |
| Figure 6 | 2-months Bad River, Step 1 (WT) | Need to reduce use cases |
| Figure 7 | 2-months Bad River, Step 2 (XWT), NWM v1.2 | Need to reduce use cases |

The authors make a clear argument that some of the traditional error metrics (e.g. RMSE, NSE) implicitly include errors in timing but don't shed much light on them. Yet, the strength of these metrics is also the simplicity of their computation. In an effort to increase the broad acceptance and use of timing metrics that are less subjective than peak-over-threshold, it might be helpful to

for the authors to attempt a recommendation for the simplest possible form of their methodology. I understand that different powers are related to independent peaks and that there is value in looking at them all. I wonder if using powers computed as the square of discharge values from traditional occurrence probability thresholds could help reconcile (connect?) the WT approach and the threshold approach. I am not suggesting more analysis here, more so a paragraph in the discussion (Section 6, around Lines 478-487) where the authors might further expand on the simplest way for others to apply their approach.

R2.5. We agree that presenting the simplest form is appealing. Some of this is related to our previous response, where we described how we evaluated what was included in the manuscript. For example, we removed the Bad River, SD, case to help focus our results on less use cases. Second, we note that we now limit our results to only looking at cluster maximums, whereas in our previous version we looked at both cluster max and cluster mean. We now note this in the Discussion and Conclusions:

> *For instance, we focus on the cluster max, but one could also examine the cluster mean. Also, instead of finding the event of maximum power in each cluster (i.e., for a given timescale), it would be possible to identify the event with maximum power in "islands of significance", i.e., significant areas contiguous in time across timescales. However, this ignores that multiple frequencies can be important at once and defining the islands when connected is problematic. If one suspected non-stationarity in the characteristic timescales over the timeseries, then a different approach such as a moving average in timescale could be employed.*

I'm not sure I fully understand the True/False (dark grey/light grey) legend in figures 5, 9, and 10. Figure 10 seems to suggest that lighter colors are not statistically significant but it's not clear from the legend. Likewise, the "Avg XWT Signif" color bar in Figures 11 and 12 is a bit mysterious to me as I see no associated colors on the graph. Could the authors rework their legends in these figures?

R2.6. We note that we have eliminated figures 9 and 10 in an effort to simplify the analysis and presentation, as mentioned above. We have left the True/False legend in Figure 5 (now Figure 6), but have clarified how XWT significance is used to determine when timing errors are "hits" in the modeled timeseries. Although this was a minor point by Reviewer 2, we note that this was also picked up by Reviewer 1, and so to this end, we have also bolstered the description of the XWT power and how it is used to diagnose if timing errors are "hits", which we include below.

We agree that we need to enhance our description of the event (dis-)agreement of observations and simulations. We address this, but first we want to remind the reviewer of the context that, in our re-organization and streamlining, we simplified our methodology and results by focusing on cluster maxs. (In the previous version we presented results for both cluster max and cluster means approaches). The cluster max is a single point of maximum power per cluster, and so cluster maxs can be classified as either hits or misses. In the previous draft we calculated the cluster mean timing error and the corresponding percent of hits within the cluster. We used the % hits in that case as a confidence measure for the timing error of each cluster. We see now how this was not clear as the hit diagnostic was different for these approaches and hence our

decision to now focus the manuscript on cluster maxs. Focusing on cluster maxs simplifies our Step 2d, which was previously called "Quantify the confidence in the timing error", which we now call "Quantify Percent Hits". Percent hits now refers to a full-timeseries diagnostic of cluster maxs (instead of individual clusters). The edited section is included below:

**3.2.4. Step 2d. Quantify Percent Hits**

*The premise of computing a timing error between the observed and modeled time series is that they share common events which can be meaningfully compared. In a two-way contingency analysis of events, a "hit" refers to when the modeled timeseries reproduces an observed event. When the modeled timeseries fails to reproduce an observed event, it is termed a "miss". In the case of a miss, it does not make sense to include the timing error in the overall assessment. Because the timing errors are calculated from the XWT, we choose to diagnose hits and misses based on the significance of the XWT. Once cluster maxima are selected based on the characteristic timescales of the observed event spectrum and timing errors are obtained at these locations in the XWT, the significance of the XWT on the cluster maxima is used to decide if the model produced a hit or a miss for each point. For a single cluster max, such as shown in Figure 3c, the XWT significance is either True or False, the point is either a hit or a miss. Table 2 also displays the results of the timing error analysis for this synthetic example. We can see the prescribed 5 hour offset and that the cluster maximum was significant in the XWT. When calculating timing errors for a longer time series, a useful diagnostic is to calculate the percent hits over all the cluster maxima in a timescale. When summarizing timing errors statistics for a timescale, we drop misses from the calculation and the % hits indicates what portion of the timeseries was dropped (% misses = 100 - % hits). In our tables we provided timing error statistics this way as well as over all observed events to reveal the impact of dropping misses.*

*Because, in step 1, we constrain our analysis to observed events in the wavelet power spectrum, we do not consider either of the remaining categories in a 2-way analysis (false alarms and correct negatives). We note that a complete 2-way event analysis could be constructed in the wavelet domain based on the Venn diagram of the observed and modeled events without necessarily using the XWT.*

The new Table 2 and Figure 3, referenced above, follow:

Table 2. Summary of timing error results for isolated peak and prescribed 5 hour offset from Onion Creek, TX, for cluster max.

| Characteristic Timescale (hr) | Avg WT Power | Number of Clusters | Cluster Max | | |
|---|---|---|---|---|---|
| | | | Timing Error (hr) | Time (hr) | Hit? |
| 22 | 598,000 | 1 | 5 | 37 | Yes |

[Figure]

*Figure 3. An isolated peak from Onion Creek, TX and a synthetic +5 hour offset: (a) observed and synthetic time series (note logged y-axis), (b) cross wavelet (XWT) power spectrum and phase angles (arrows), (c) sampled timing errors for observed events (dashed contour is XWT significant events) and star is cluster maximum.*

Related to the decision to focus on cluster maxs and their diagnosis as hits or misses using the XWT, earlier in the manuscript (Step 2a) we have also edited the cross wavelet (XWT) figure panels to show this visually. This is seen in Figure 3 (previously Figure2, Onion Creek synthetic example) above and in its caption. Specifically, we have adjusted panel c, to show how the observed events (colors) don't exactly overlap with the XWT significant events (dashed contour). This is now explained in the methodology:

> *Similar to Step 1b of the WT, we can also calculate areas of significance for the XWT power as shown by the black contour in Figure 3b. For the XWT, significance is calculated with respect to the theoretical background wavelet spectra of each timeseries (Torrence and Compo, 1998). We define XWT events as points of significant XWT power outside the COI. XWT events indicate significant joint variability between the observed*

*and modeled timeseries. In the next section, we employ XWT events as a basis for identifying hits and misses on observed events for which the timing errors are calculated. Described further in step 2d, timing errors are valid only for hits. Figure 3c shows the intersection of the observed events (colors) and the XWT events (dashed contour). This is a region of hits. Note that the early part of the observed events, particularly at shorter timescales, is not in the XWT events. This is because of the timing offset in the modeled timeseries, which misses the early part of the observed event.*

For Figures 11 and 12 (now 10 and 11) we have renamed the color scale scale "Percent Hits" and clarified the caption to include: "... outline shading shows percent (%) hits in the cross wavelet transform (XWT)." This is shown for Figure 11, below:

[Figure]

*Figure 11. Five year run from Pemigewasset River, NH: Comparing cluster max timing error distributions for top three characteristic timescales (see panel title) across NWM versions; outline shading shows percent (%) hits in the cross wavelet transform (XWT).*

In our original tables, timing errors were calculated over all observed events (i.e., both hits and misses) and we reported the percent hits (previously called "Avg % Significant in XWT"). We will calculate timing errors over only the hits for each timescale and add this as an additional column

to the tables. We may drop the original timing error statistic over all observed events (hits and misses), but for now we plan to compare the two.

**Technical corrections**

 Line 296-297: missing "Land" in the acronym for NLDAS.
This has been fixed.

---

## Author Response (AR1)

**Response to Reviewers**

**General Response**

We thank Reviewer 1 and Reviewer 2 for their detailed reviews, and for sharing their constructive and insightful comments. Before our point-by-point response, we note several major changes in the manuscript organization and content based on comments from both reviewers. We have addressed all review comments in our responses as well as in a substantial revision to the manuscript. The resulting manuscript is clearer and much improved.

In terms of manuscript organization, we have consolidated the methodology to be one section. Section 3 now combines previous sections 2, 3, and 4.2. This improves the clarity of the method and reduces redundancies. In terms of content, we have carefully re-evaluated what use cases, figures, and tables to include in the paper. For example, we have removed one of the locations, Bad River, SD, from the Results entirely (Section 4). In doing so, we have culled our figures and tables such that all of the figures and tables are in the main manuscript. We currently have one less figure than the last draft (11 Figures now, versus 12 Figures in last draft), and we no longer have a Supplemental document. This streamlining has helped to simplify the paper and to improve its clarity. One notable example of this is that we now focus the methodology and results on cluster maxima, whereas in the previous version we included methods/results from cluster maxima and cluster means. By focusing on the cluster maxima analysis, we clarify how the (dis)agreement between the events in the observed WT and modeled XWT can be used to quantify "hits/misses". Further, we now only report timing errors for hits in the timing error statistics.

In summary, we thank the reviewers for bringing these points to our attention. As a result of their careful reviews, we have added numerous clarifications to the text and figures in an effort to improve overall understanding and interpretation. Below, we provide a point-by-point response, where we address each of Reviewer 1 and Reviewer 2's concerns, respectively.

**Reviewer 1.**
Dear Editor, dear Authors, Please find my comments in the attachment. Yours sincerely, Uwe Ehret

Review of Manuscript 'A Wavelet-Based Approach to Streamflow Event Identification and Modeled Timing Error Evaluation' (hess-2020-323) by E. Towler and J. L. McCreight Dear Editor, dear Authors,

I have reviewed the aforementioned work. My conclusions and comments are as follows:

**1. Scope**

The article is within the scope of HESS.

**2. Summary**

The authors explain a method based on wavelet transform and cross wavelet transform to i) detect relevant events in streamflow time series and ii) to calculate timing errors between a reference time series for which the events were determined (typically an observed time series) and a test time series (typically a model simulation). Relevant regions in the full 2-d space (time and timescale) of wavelet transforms are identified by significance testing as suggested by Torrence and Compo (1998). The timing errors are calculated based on the cross wavelet transform as suggested by Liu et al. (2011), but restricted to the areas of significant events in the reference time series, which imposes a direction on the comparison. The authors illustrate their method with several application examples, which vary by their event characteristics (single event, multiple events, events caused by different processes). The authors conclude that the proposed method offers a systematic, objective, and data-driven method for event identification and timing error calculation, which can be applied to large data sets, and they stress that beyond the particular application (including particular user choices) presented in the paper, other uses of the core method are possible.

**3. Overall ranking**

Overall, the authors provide an elegant and general solution to the fuzzy problem of event detection and timing error calculation in streamflow time series, which I am sure provides more generality, better reproducibility and better insight than most existing methods, including the ones I was involved with (Series Distance). There are, however, some flaws in the study, in terms of presentation clarity and in terms of demonstrating the generality of the method beyond the particular chosen use case, which should be eliminated. The relevance of the method deserves this extra effort.

We truly appreciate your careful review and thoughtful suggestions. Please see our point-by-point responses, below.

**4. Evaluation**

General points

In the introduction, please provide a more comprehensive literature review on methods for event detection and timing error calculation. E.g. Mei and Anagnostou (2015), Merz et al. (2006), Koskelo et al. (2012).

Thank you for this suggestion and these citations. We have added these references to expand our introduction to event detection to include baseflow separation methods. The new excerpt from the Introduction is provided here (changes in **bold**, line 40+):

The fundamental challenge with evaluating timing errors is identifying what constitutes as an "event" in the two time series being compared. Identifying events is typically subjective, time consuming, and not practical for large-sample hydrological applications (Gupta et al. 2014). A variety of baseflow separation methods, ranging from physically-based to empirical, have been developed to identify hydrologic events (see Mei and Anagnostou 2015 for a summary), though many of these approaches require some manual inspection of the hydrographs. Merz et al. (2006) put forth an automated approach, but it requires a calibrated hydrologic model, which is a limitation in data poor regions. Koskelo et al. (2012) developed a simple, empirical approach that only requires rainfall and runoff time series, but is limited to small watersheds and daily data. Mei and Anagnostou (2015) introduce an automated physically-based approach, which is demonstrated for hourly data, though one caveat is that basin events need to have a clearly detectable recession period. Additional methods for identifying events have focused on identifying flooding events...

The use case presented in the paper takes observed events as the reference, and calculates timing errors for these (see e.g. P4 L90-92). This neglects other important aspects of event-specific (dis-)agreement of observations and simulations: False alarms and missed events. This is mentioned several times by the authors (e.g. P7 L148-153, P12 L266, P12 L271-272, P17 L388-391), and they also mention that the method could be set up differently if these aspects are of interest, but they do not explain how. False alarms and missed events are important and often-used features of categorical model evaluation (and the idea of 'event' is categorical). So I suggest that the authors add a short discussion about if and how their method can be used to measure them (I am not asking to actually perform these analyses, but to provide guidance for future uses).

We agree that we need to enhance our description of the event (dis-)agreement of observations and simulations. We address this, but first we want to remind the reviewer of the context that, in our re-organization and streamlining, we simplified our methodology and results by focusing on cluster maxima. (In the previous version we presented results for both cluster maxima and cluster mean approaches. Note that we also clarify the definition of cluster, please see later responses.) The cluster maxima are the single points of maximum power per cluster, and so cluster maxima can be classified as either hits or misses. In the previous draft we also calculated the cluster mean timing error and the corresponding percent of hits within the cluster. We used the % hits in that case as a confidence measure for the timing error of each cluster. We see now how this was not clear as the hit diagnostic was different for these approaches and hence our decision to now focus the manuscript on cluster maxima. Focusing on cluster maxima simplifies our Step 2d, which was previously called "Quantify the confidence in the timing error", which we now call "Filter Misses". Percent hits now refers to a full-time series diagnostic of cluster maxima (instead of individual clusters). The edited section is included below (line 327+):

**3.2.4. Step 2d. Filter Misses**

The premise of computing a timing error between the observed and modeled time series is that they share common events which can be meaningfully compared. In a two-way contingency analysis of events, a "hit" refers to when the modeled time series reproduces an observed event. When the modeled time series fails to reproduce an observed event, it is termed a "miss". In the case of a miss, it does not make sense to include the timing error in the overall assessment. Once the characteristic timescales of the observed event spectrum are identified and event cluster maxima are located, timing errors are obtained at these locations in the XWT. In this step, the significance of the XWT on these event cluster maxima is used to decide if the model produced a hit or a miss for each point and to determine if the timing error is valid. As previewed above, Figure 3c shows the observed events (colors) and the dashed contour shows intersection between the observed and XWT events. Regions of intersection between observed events and XWT events are considered model hits and observed events falling outside the XWT events are considered misses. Because we constrain our analysis to observed events in the wavelet power spectrum, we do not consider either of the remaining categories in a 2-way analysis (false alarms and correct negatives). We note that a complete 2-way event analysis could alternatively be constructed in the wavelet domain based on the Venn diagram of the observed and modeled events without necessarily using the XWT. We choose to use the XWT events because the XWT is the basis of the timing errors.

In the synthetic example of Onion Creek, a single characteristic timescale and event cluster yields a single cluster maximum as shown by the star in Figure 3c. Because this star falls both within the observed and XWT events, it is a hit and the timing error at that point is valid (Table 2). For a longer time series, as seen in subsequent examples, a useful diagnostic and compliment to timing error statistics at each characteristic timescale is the percent hits. When summarizing timing error statistics for a timescale, we drop misses from the calculation and the % hits indicates what portion of the time series was dropped (% misses = 100 - % hits). In our tables we provided timing error statistics only for hits.

The new Table 2 and Figure 3, referenced above, follow:

Table 2. Summary of timing error results for isolated peak and prescribed 5 hour offset from Onion Creek, TX, for cluster maxima analysis.

| Characteristic | Ανσ WT  | WT Number of
er Clusters | Cluster Maxima       |           |      |
|----------------|---------|-----------------------------|----------------------|-----------|------|
| Timescale (hr) | Power   |                             | Timing Error
(hr) | Time (hr) | Hit? |
| 22             | 555,700 | 1                           | 5                    | 37        | TRUE |

---

## Author Response (AR2)

Response to Reviewers

Referee # 1:

Dear Authors,

Congratulations on the improvements of your paper, it reads much better now. Two minor
things remain:
Siebert --> Seibert (p3)
Section 6 --> Section 5

Best regards, Uwe Ehret

Thank you for your review. We have made the two minor corrections you suggest in the final
manuscript.